biomathematics/health and disease and epidemiology/mathematical modelling

Bayesian inference, epidemiology, COVID-19

**Author for correspondence:**
Robert L. Jack
e-mail: rlj22@cam.ac.uk

# Efficient Bayesian inference of fully stochastic epidemiological models with applications to COVID-19

Yuting I. Li[1], Günther Turk[1,†], Paul B. Rohrbach[1,†], Patrick Pietzonka[1,†], Julian Kappler[1,†], Rajesh Singh[1,†], Jakub Dolezal[1,†], Timothy Ekeh[1], Lukas Kikuchi[1], Joseph D. Peterson[1], Austen Bolitho[1], Hideki Kobayashi[2], Michael E. Cates[1], R. Adhikari[1] and Robert L. Jack[1,2]

[1]Department of Applied Mathematics and Theoretical Physics, University of Cambridge, Wilberforce Road, Cambridge CB3 0WA, UK
[2]Yusuf Hamied Department of Chemistry, University of Cambridge, Lensfield Road, Cambridge CB2 1EW, UK

 PBR, 0000-0001-6240-6872; JK, 0000-0002-8559-907X;
RS, 0000-0003-0266-9691; RLJ, 0000-0003-0086-4573

Epidemiological forecasts are beset by uncertainties about the underlying epidemiological processes, and the surveillance process through which data are acquired. We present a Bayesian inference methodology that quantifies these uncertainties, for epidemics that are modelled by (possibly) non-stationary, continuous-time, Markov population processes. The efficiency of the method derives from a functional central limit theorem approximation of the likelihood, valid for large populations. We demonstrate the methodology by analysing the early stages of the COVID-19 pandemic in the UK, based on age-structured data for the number of deaths. This includes maximum *a posteriori* estimates, Markov chain Monte Carlo sampling of the posterior, computation of the model evidence, and the determination of parameter sensitivities via the Fisher information matrix. Our methodology is implemented in PyRoss, an open-source platform for analysis of epidemiological compartment models.

## 1. Introduction

The ongoing COVID-19 pandemic has demonstrated the vital importance of epidemiological forecasting [1–8]. Given the large

†Contributed equally.

uncertainties in the mechanisms of viral transmission, and the difficulties in determination of numbers of infections and deaths, a Bayesian approach is natural [9–14]. This allows the range of likely outcomes to be quantified and characterized. The evidence in favour of different epidemiological models can also be assessed, in the light of data.

Compartment models are widely used as models of epidemiological dynamics [15–17]. Within these models, individuals are grouped into cohorts, for example according to their age or location. The key assumption is that the rates of contact between individuals depend only on their cohorts. The resulting models have sufficient complexity to be useful in forecasting, while remaining simple enough that Bayesian analyses are tractable [10,11,13,18].

Such analyses require three main ingredients: the definition of a model, the prior distributions of the inference parameters and an efficient method for the evaluation of the posterior distribution [15,19–23]. In this work, we derive an approximation to the model likelihood directly from the model definition, via a functional central limit theorem (CLT), similarly to [24–30]. Hence, for any given model, the approximated likelihood can be derived by a generic and automated procedure. This enables rapid Bayesian fitting of models to data with fully quantified uncertainties, as implemented in the PyRoss package [31]. It also enables sampling from the posterior by Markov chain Monte Carlo (MCMC), and the evaluation of model evidence (also known as marginal likelihood), which enables Bayesian model comparison [32–34]. The results presented here build on an earlier technical report [13] which discussed automated fitting of such models to data.

A variety of Bayesian inference approaches are possible in calculations of this type, which make different assumptions (either implicit or explicit) about the role of random fluctuations in the disease propagation and the surveillance of the epidemic. A common approach is to consider a deterministic generative model for the disease, and to treat the data collection (surveillance) as a stochastic process [9,11,18,35]. The disease dynamics is analysed by solving ordinary differential equations (or equivalent equations in discrete time); the likelihood is then computed by a simple formula. This approach is fast and flexible, but the use of deterministic disease models can bias the results, as can assumptions about independence of observed data points: see for example [36]. Other approaches [37–39] consider fully stochastic compartment models and estimate parameters using particle filters (or sequential Monte Carlo methods). Such computations avoid the biases mentioned above, but are much more expensive (for any given parameter set, multiple stochastic trajectories must be generated, and one aims to optimize over all parameter choices).

The methodology that we present is intermediate between these two approaches: the aim is to mitigate the biases associated with deterministic disease models, without the computational cost of stochastic simulations. The CLT approximation to the likelihood can be evaluated quickly by solving ordinary (deterministic) differential equations [25–30]. The underlying models include stochastic aspects of disease transmission, and the approach avoids any assumption of independent data points. On the other hand, the CLT approximation assumes that the epidemic is spreading in a large well-mixed population. As such, it can suffer from bias if applied to localized outbreaks or small populations. In such cases, the CLT approximation is no longer suitable, but methods for computing the likelihood from stochastic simulation should be applicable [37–39].

As an example where the proposed methodology is appropriate, we analyse an age-resolved population-level model of England and Wales, using data for recorded deaths from COVID-19 over the period 6 March to 15 May 2020, and inferring more than 40 model parameters, with priors informed by existing literature. Given the large numbers of cases in this period, the CLT approximation to the likelihood is justifiable. We compare several variants of the model, which differ in their assumed contact structure; we also compare the model evidence [32–34] for the different variants. For such large models (with so many parameters), methods that estimate the likelihood by simulation of stochastic trajectories are intractable. To our knowledge, previous work on epidemiological inference within CLT approximations [26–30] have not analysed models of this complexity.

A more detailed picture of the epidemic would be available by combining multiple data sources (for example, positive tests as well as deaths), but the example presented here illustrates the general methodology. The intended future applications of these methods are to similar (population-level) models with higher complexity, e.g. [40].

In the following, models and definitions are given in §2, the likelihood approximation is discussed in §3, and the inference methods are summarized in §4. The approach is validated in §5 by performing inference on a synthetic dataset for a simple compartment model. The models for England and Wales

are defined in §6, while §7 shows the results. We conclude with a discussion in §8. Some technical details and supplementary results are provided in appendices.

# 2. Compartment models

## 2.1. Definition

Consider a compartment model where $N$ individuals are grouped into $M$ cohorts, according to some attributes (for example, age and/or gender). Each cohort is divided into $L$ epidemiological classes, indexed by $\ell = 1, 2, \ldots, L$. We assume a single susceptible class, which is $\ell = 1$. Other classes may be either infectious or non-infectious: the canonical example is an SIR model which corresponds to $L = 3$, in which case the recovered (R) class is non-infectious. The analysis presented here is straightforwardly generalized to more complex compartment models, as might be used (for example) to model different pathogen strains, or vaccinated individuals with reduced susceptibility, or testing and quarantining [40].

In the general case, the total number of compartments is $M \times L$ and the state of the system can be specified as a vector

$$\boldsymbol{n} = (n_1, n_2, \ldots, n_{M \times L}). \tag{2.1}$$

We use boldface notation throughout this work to indicate both vectors and matrices. Each element of $\boldsymbol{n}$ is a non-negative integer, such that the number of individuals in class $\ell$ and cohort $i$ is $n_{i+M(\ell-1)}$. For example, $n_1, \ldots, n_M$ are the number of susceptible individuals in each cohort.

The disease propagation involves individuals moving between the epidemiological classes, by a Markov population process [41]. (Models may also include immigration or emigration steps where the total population changes.) The parameters of the model are $\boldsymbol{\theta} = (\theta_1, \theta_2, \ldots)$, indexed by a label $a$. The various stochastic transitions are indexed by $\xi = 1, 2, \ldots$. In transition $\xi$, the population $\boldsymbol{n}$ is updated by a vector $\boldsymbol{r}_\xi$ with integer elements, that is

$$\boldsymbol{n} \to \boldsymbol{n} + \boldsymbol{r}_\xi \quad \text{with rate } w_\xi(t, \boldsymbol{\theta}, \boldsymbol{n}). \tag{2.2}$$

For example, if transition $\xi$ involves a single individual moving from compartment $\alpha$ to compartment $\beta$ then $\boldsymbol{r}_\xi$ has $-1$ in the $\alpha$-th place and $+1$ in the $\beta$-th place, with all other elements being zero. Consistent with the Markovian assumption, the rate $w_\xi(t, \boldsymbol{\theta}, \boldsymbol{n})$ depends on the current state, the parameters of the model and the time $t$.

Two common types of transition are infection, and progression from one stage to another. For example, in the simple SIR example (with $M = 1$), we write $(n_1, n_2, n_3) = (S, I, R)$ with total population $N = S + I + R$. Taking infection and recovery parameters as $\boldsymbol{\theta} = (\beta, \gamma)$, the infection transition has $\boldsymbol{r}_1 = (-1, 1, 0)$ and rate $w_1 = \beta SI/N$, while progression for $I$ to $R$ has $\boldsymbol{r}_2 = (0, -1, 1)$ and $w_2 = \gamma I$. The general formalism used here covers simple SIR models as well as more complex ones, e.g. §6.

## 2.2. Contact dynamics and the well-mixed assumption

As illustrated by the SIR example, it is a general feature that progression transitions have rates that are linear in $\boldsymbol{n}$, but infections are bilinear. As usual, we consider compartment models that assume a well-mixed population, in the sense that the typical frequency of meetings between individuals depends only on their cohort. These frequencies are described by the *contact matrix* [15–17,42], which appears in the rates $w_\xi$ for infectious transitions (for an example, see §6 below).

This well-mixed assumption neglects the detailed social structure of the population, for example that friends and family members meet each other much more frequently than other individuals. Despite this (coarse) approximation, compartment models are valuable tools for practical analysis of epidemics, and are useful for inference. Still, it must be borne in mind in the following that these models are not microscopically resolved descriptions of individuals' behaviour, but rather approximate descriptions that capture the main features of disease dynamics, and its dependence on model parameters.

## 2.3. Average dynamics and law of large numbers

We will be concerned with epidemics in large populations, with the well-mixed assumptions described above. We consider an approximate likelihood that is derived by considering a limit of large population,

which is controlled by a large parameter $\Omega$. In models where the total population $N$ is fixed then $\Omega = N$. If the population is uncertain or subject to change then $\Omega$ is taken as a suitable reference value, for example the prior mean population at time $t = 0$. (As an example with changing population, we imagine a model that includes birth of new individuals and death by non-epidemiological causes, which can be modelled by transitions that add/remove individuals to/from $S$ (or other) classes, instead of transferring them between classes.)

Now define

$$x = \frac{1}{\Omega}n, \tag{2.3}$$

whose elements indicate the fractions of individuals in each compartment. To ensure a suitable large-population limit (within the well-mixed assumption), we require that the rates $w_\xi$ have a specific dependence on $\Omega$

$$w_\xi(t, \boldsymbol{\theta}, \Omega x) = \Omega \omega_\xi(t, \boldsymbol{\theta}, x), \tag{2.4}$$

where $\omega_\xi$ is the transition rate per individual (as opposed to the rate for the population). This assumption corresponds to frequency-dependent transmission, as is commonly assumed in models of human disease [17]. The methodology described here can be generalized to models with density-dependent transmission, but we focus here on the frequency-dependent case, which is the relevant one for application to COVID-19.

Given the parameters $\boldsymbol{\theta}$ and an initial condition $x(0)$, models of this form obey a law of large numbers in the limit of large population $\Omega \to \infty$ [43–46]. In this limit, almost all stochastic trajectories $x(t)$ lie close to a single deterministic trajectory, $\bar{x}(t)$, which can be obtained as the solution of an ordinary differential equation

$$\frac{\mathrm{d}\bar{x}}{\mathrm{d}t} = \sum_\xi r_\xi \omega_\xi(t, \boldsymbol{\theta}, \bar{x}). \tag{2.5}$$

The sum in this equation runs over all possible values of $\xi$; we do not write the range explicitly in such cases, for compactness of notation. Equation (2.5) is straightforwardly solved by numerical methods, so $\bar{x}$ can be computed.

Note that the initial condition for (2.5) is $x(0)$. As $\Omega \to \infty$, this means that a finite fraction of the population must be infected at $t = 0$, which is required for the law of large numbers to hold. As a result, this theory does not apply in the very early stages of an epidemic where only a few individuals have been infected.

## 2.4. Central limit theorem

The (approximate) likelihood that we use for Bayesian analysis rests on a functional CLT [43–47] for fluctuations of the epidemiological state about the mean value $\bar{x}$. The structure of the CLT is outlined here, it applies in the limit $\Omega \to \infty$. The associated approximation for likelihood is discussed in §3.2, which also discusses its applicability when $\Omega$ is finite.

The CLT is derived for a fixed initial condition $x(0)$. To analyse fluctuations, consider the (scaled) deviation of the epidemiological state $x$ from its average

$$u(t) = \sqrt{\Omega}[x(t) - \bar{x}(t)], \tag{2.6}$$

with $u(0) = 0$. (The factor of $\sqrt{\Omega}$ is standard in CLTs, it is chosen so that typical trajectories of the model have $u$ of order unity, as $\Omega \to \infty$.) By considering the increment in $u$ over a short time-interval and taking the limit of large $\Omega$, one finds [48, §4.5.9] that $u$ obeys a stochastic differential equation

$$\mathrm{d}u = J(t, \boldsymbol{\theta}, \bar{x})u\,\mathrm{d}t + \sum_\xi \boldsymbol{\sigma}_\xi(t, \boldsymbol{\theta}, \bar{x})\,\mathrm{d}W_\xi, \tag{2.7}$$

where $W_1, W_2, \ldots$ are independent standard Brownian motions (Wiener processes); our notation suppresses the dependence of $\bar{x}$ and $u$ on the time $t$, for compactness. The elements of the square matrix $J$ are

$$J_{ij}(t, \boldsymbol{\theta}, \bar{x}) = \sum_\xi r_{\xi i} \frac{\partial}{\partial x_j} \omega_\xi(t, \boldsymbol{\theta}, x)\bigg|_{x=\bar{x}}, \tag{2.8}$$

where $r_{\xi,i}$ is the $i$th element of the vector $r_\xi$. Similarly,

$$\boldsymbol{\sigma}_\xi(t, \boldsymbol{\theta}, \bar{x}) = \boldsymbol{r}_\xi \sqrt{\omega_\xi(t, \boldsymbol{\theta}, \bar{x})}. \tag{2.9}$$

In the physics literature, the derivation of (2.7) uses the van-Kampen expansion [46,48]; the application in population dynamics is due to Kurtz [43–45].

One sees that $J$ and $\boldsymbol{\sigma}_\xi$ depend on the deterministic path $\bar{x}$ but not on the random variable $u$, so (2.7) is a time-dependent Ornstein–Uhlenbeck process. The CLT applies as $\Omega \to \infty$, it states that $u$ has Gaussian fluctuations with mean zero, and a covariance that can be derived from (2.7). This result applies to the covariance at any fixed time, and to correlations between fluctuations at different times. The correlations are discussed in appendix A, for example (A 6, A 8).

# 3. Data and likelihood

A central task in Bayesian inference is to compute the posterior distribution of the parameters $\boldsymbol{\theta}$, given some observational data. The posterior probability density function (pdf) of the parameters is [19–21]

$$P(\boldsymbol{\theta}|\text{data}) = \frac{P(\text{data}|\boldsymbol{\theta})P(\boldsymbol{\theta})}{Z(\text{data})}, \tag{3.1}$$

where $Z(\text{data})$ is called the model evidence, which is fixed by normalization of the posterior.

We now describe how the observed data are incorporated in our methodology, after which we discuss the likelihood. A technical aspect of our approach is that the initial condition of the system at time $t = 0$ must be parametrized in terms of $\boldsymbol{\theta}$ (or explicitly provided).

## 3.1. Data

In practical situations, observations of the epidemiological state are subject to uncertainty. Our methodology includes all random aspects of the observation (surveillance) process directly into the model. This means that measured data can be identified with the populations of certain model compartments; the remainder of the compartments are not observed, and correspond to latent variables. For example, the model of §6 (below) includes an observed compartment for deceased individuals, which should properly be interpreted as a compartment for deceased individuals who were diagnosed with COVID-19. Other compartments—for example susceptible and infected—are latent variables, which are independent of diagnosis. The measurement process is then modelled through the (stochastic) transition from infected compartment to deceased compartment, whose rate depends on the probability of correct diagnosis.

We assume that observations are made at an ordered set of positive times, indexed by $\mu = 1, 2, \ldots$. Specifically, at time $t_\mu$, one observes a vector with $m_{\text{obs}}$ elements, which are linear combinations of the compartment populations at that time. That is,

$$\boldsymbol{n}^{\text{obs}}(t_\mu) = \boldsymbol{F}\boldsymbol{n}(t_\mu), \tag{3.2}$$

where $F$ is a matrix of size $m_{\text{obs}} \times (ML)$, which we call the *filter matrix*.

Now define a vector $Y$ that contains all the observed data, by collecting the individual observation vectors,

$$\boldsymbol{Y} = \left(\boldsymbol{n}^{\text{obs}}(t_1), \boldsymbol{n}^{\text{obs}}(t_2), \ldots\right). \tag{3.3}$$

This vector corresponds to the data in (3.1).

## 3.2. Approximated likelihood

The likelihood is denoted by

$$\mathcal{L}(\theta) = P(\boldsymbol{Y}|\boldsymbol{\theta}). \tag{3.4}$$

Hence we require a computationally tractable estimate of this probability. The formula that we use is based on the CLT for the path $x(t)$, as discussed in §2.4.

Given the parameters $\boldsymbol{\theta}$, the most likely observation is $\overline{Y}$, whose elements are

$$\overline{\boldsymbol{n}}^{\mathrm{obs}}(t_\mu) = \Omega F \overline{\boldsymbol{x}}(t_\mu), \tag{3.5}$$

recall (2.3), (3.2), (3.3). Define also the (scaled) deviation of the data from this value,

$$\boldsymbol{\Delta} = \frac{\boldsymbol{Y} - \overline{\boldsymbol{Y}}}{\sqrt{\Omega}}. \tag{3.6}$$

As in (2.6), the scaling is such that elements of $\boldsymbol{\Delta}$ are typically of order unity. In the specific case considered here, the functional CLT states that the log-likelihood obeys

$$\log \mathcal{L}(\theta) \simeq -\frac{1}{2}\left[\boldsymbol{\Delta}^T \boldsymbol{G}^{-1} \boldsymbol{\Delta} + \ln\det\left(\frac{2\pi \boldsymbol{G}}{\Omega}\right)\right], \tag{3.7}$$

where the approximate equality is accurate as $\Omega \to \infty$, and $\boldsymbol{G}^{-1}$ denotes the inverse of a square covariance matrix $\boldsymbol{G}$, whose form is dictated by the CLT of §2.4: see appendix A, in particular (A 8). Given a compartment model with parameters $\boldsymbol{\theta}$ and a filter matrix $F$, computation of $\boldsymbol{G}$ requires numerical solution of a (matrix-valued) ODE.

We note once more that (3.7) is an approximation to the likelihood of the underlying compartment model, valid for large populations. Similar approximations have been applied previously in epidemiological inference, for example [26–30], and in physical sciences [25,49,50]. The results of those studies indicate that inference based on the CLT approximation can be effective in practice, but for any given $\Omega$, the accuracy of the Gaussian (CLT) assumption is not easy to assess.

To address this, we highlight a few situations where caution is advised, in practical settings. First, the CLT is restricted to typical fluctuations of the stochastic process, so its application requires that the observed data $Y$ lie within a few standard deviations of their most likely values $\overline{Y}$. In other words, the likelihood will be only accurate for models with reasonable fit to the data. Second, for computation of the mean trajectory, the error in the CLT approximation comes from nonlinear processes (such as infection of a susceptible individual), while linear processes (such as recovery of an infectious individual) do not require any approximation. For well-mixed models, this means that a necessary condition for the CLT is that the *total* number of infectious individuals should be large compared with unity, so that the fluctuations in this quantity are not too large. Third, changes in compartment populations are integers, but they are treated as real numbers by the CLT. The associated error is that of replacing a (difference of) Poisson-distributed integers by a Gaussian-distributed real number.

Among these three factors, the first two must be taken seriously when applying the methodology proposed here. In the examples considered below, the models do fit the data, and the total number of infectious individuals is numerically large at all times considered, giving confidence in the CLT approximation. For the third factor, we note that if some compartment populations are numerically small, observations of these compartments will tend to have little impact on the likelihood (because their mean occupancies will probably be comparable with their variances). In this case, the methodology will correctly infer that this observation has little effect on the posterior distribution: this weak dependence can help to mitigate errors associated with a breakdown of the CLT. We return to this point in §5, below.

Finally, we observe that in the practical context of epidemiology, the approximation error of the likelihood must be considered together with the fact that any compartment model is already a coarse approximation of the real-world disease progression. This is especially true for population-level models, given the well-mixed assumption for contacts within cohorts. In such cases, the aim is not for absolute accuracy in parameter estimation or likelihood computation. Instead, the likely applications would be Bayesian model comparison and forecasting, as discussed in §4. For those applications, it is vital to address sources of systematic bias in the inference process. By incorporating stochastic disease progression and dependency among observed data points, the CLT mitigates at least some of the biases of simpler approaches [36], at manageable computational cost.

# 4. Inference methodology

This section briefly describes the inference methodology, as implemented in PyRoss [13,31].

## 4.1. Model estimation

The methodology has been implemented for a general class of compartment models as defined above, including progression and infection transitions. Specifically, if transition $\xi$ involves progression from compartment $\alpha$, one has

$$w_\xi(t, \boldsymbol{\theta}, \boldsymbol{n}) = \gamma_\xi(t, \boldsymbol{\theta}) n_\alpha, \tag{4.1}$$

with arbitrary dependence of $\gamma_\xi$ on the parameters $\boldsymbol{\theta}$ and the time $t$. For infection reactions, suppose that transition $\xi$ involves a susceptible individual in cohort $i$ being infected by an individual in some infectious class. Denote the population of susceptible individuals in cohort $i$ by $S_i$ and the population of individuals in cohort $j$ of the infectious class by $I_j^{(k)}$; here $k$ is a label for the relevant infectious class. Then the generic infection rate is

$$w_\xi(t, \boldsymbol{\theta}, \boldsymbol{n}) = \sum_{j=1}^{M} K_\xi(t, \boldsymbol{\theta}) \frac{S_i I_j^{(k)}}{\Omega}, \tag{4.2}$$

where $K_\xi$ has arbitrary dependence on the parameters $\boldsymbol{\theta}$ and the time $t$. The form of $K$ depends on the rates of contacts between cohorts and on various epidemiological parameters, a specific example is given in §6 below.

Once the model is specified, the inference methodology is automated. We outline the method, with details in appendix B and [13]. Given the data and some parameter values $\boldsymbol{\theta}$, the (non-normalized) posterior is computed (up to the normalization factor $Z$) by combining the prior information with (3.7). This posterior is optimized over $\boldsymbol{\theta}$ using the covariance maximization evolutionary strategy (CMA-ES) [51], yielding the maximum *a posteriori* (MAP) parameters $\boldsymbol{\theta}^*$. We also compute the Hessian matrix of the log-posterior using finite differences.

We consider the Fisher information matrix (FIM) [19], which measures the information provided by the data about the inferred parameters of the model. It is a matrix with elements

$$\mathcal{I}_{ab}(\boldsymbol{\theta}) = -\left\langle \frac{\partial^2}{\partial \theta_a \partial \theta_b} \log \mathcal{L}(\boldsymbol{\theta}) \right\rangle, \tag{4.3}$$

where the angled brackets denote an average over the stochastic dynamics of the model, with fixed parameters $\theta$. Recalling (3.4), this means that one averages over all possible values of $Y$ according to the model dynamics, instead of using the observed data. The sensitivity of parameter $a$ with respect to the (expected) data can then be estimated as

$$s_a = \theta_a^* \sqrt{\mathcal{I}_{aa}(\boldsymbol{\theta}^*)}, \tag{4.4}$$

for more detail see [52,53]. The FIM is defined as an average over the stochastic dynamics, but the Gaussian structure of the likelihood (3.7) means that the FIM can be estimated by a deterministic computation, see appendix B.1.

## 4.2. Posterior sampling and the role of priors

To go beyond the MAP, we sample the posterior for $\boldsymbol{\theta}$ by MCMC, using the emcee package [54]. In what follows, the results depend significantly on the prior, as well as the likelihood. This is natural in our (Bayesian) approach, because there are many sources of uncertainty in epidemiological modelling, and we incorporate available knowledge into prior distributions, informed by whatever expert judgement is available. Posterior sampling reveals which parameters are identifiable (constrained by the data) and which are only weakly identifiable (their posterior distribution remains close to the prior). The result of this process is that identifiable parameters are determined by the data, while weakly identifiable ones are determined by expert judgement, through the prior. (For experiments in the physical sciences, one might hope for enough data that the inferred parameters depend weakly on the prior, but that is unlikely in the epidemiological context.)

## 4.3. Model comparison

A significant advantage of Bayesian approaches is the ability to compare the evidence for different models in the light of data [32–34]. Several criteria exist for choosing among different models. We

consider here the model evidence: this is not as easy to compute as some other criteria, but it has a firm theoretical basis, see for example ch. 28 of [21].

Recalling (3.1), the evidence in favour of any model may be expressed in terms of the likelihood $\mathcal{L}$ and the prior $P$ as

$$Z = \int \mathcal{L}(\boldsymbol{\theta})P(\boldsymbol{\theta})\,\mathrm{d}\boldsymbol{\theta}, \tag{4.5}$$

which is also known as the marginal likelihood. It tends to be large if the model corresponds to high likelihood, but the integral over parameters $\boldsymbol{\theta}$ means that $Z$ is strongly suppressed in cases where fitting the data requires fine-tuning of the parameters. This ensures that overfitted models have low evidence. Hence, models (or hypotheses) with larger $Z$ are to be preferred (at least in the absence of prior information about which model is more likely). In practice, it is more convenient to work with the log-evidence.

We compute $Z$ using thermodynamic integration, see appendix B.2. The evidence is useful for Bayesian model comparison and model averaging [33], an example of model comparison is given in §7.3 below.

To interpret the model evidence, it is also useful to compute the deviance $\overline{D}$ [55], which is related to the posterior average of the log-likelihood as $\overline{D} = -\mathbb{E}_{\mathrm{post}}[\log \mathcal{L}]$. Noting that the posterior distribution is $P_{\mathrm{post}}(\boldsymbol{\theta}) = \mathcal{L}(\boldsymbol{\theta})P(\boldsymbol{\theta})/Z$ one has

$$\overline{D} = - \int P_{\mathrm{post}}(\boldsymbol{\theta}) \log \frac{Z P_{\mathrm{post}}(\boldsymbol{\theta})}{P(\boldsymbol{\theta})}\,\mathrm{d}\boldsymbol{\theta}. \tag{4.6}$$

This may be rearranged as

$$\log Z = -\overline{D} - \mathcal{D}_{\mathrm{KL}}(P_{\mathrm{post}}\|P), \tag{4.7}$$

where $\mathcal{D}_{\mathrm{KL}}(P_{\mathrm{post}}\|P)$ is the Kullback–Leibler (KL) divergence between prior and posterior. Hence, the evidence is large for models with high likelihood (low deviance), but subtracting the KL divergence means that the evidence is penalized for models where the posterior distribution is too sharply peaked, or too different from the prior assumptions. This avoids overfitting [34].

## 4.4. Forecasts and nowcasts

Given samples from the posterior, several kinds of forecast and nowcast are possible. The time period over which data is used for inference is called the *inference window*.

In a *deterministic* forecast or nowcast, we compute the average path $\overline{x}(t)$ for a given set of parameters. This allows prediction of the population of unobserved (latent) compartments. If this is performed for times $t$ within the inference window, we refer to it as a nowcast. The path $\overline{x}(t)$ can also be computed outside this window, this is a forecast. By sampling parameters from the posterior, the range of behaviour can be computed. However, this computation only captures the role of parameter uncertainty, it neglects the inherent stochasticity of the model.

In a *conditional* nowcast, we use the functional CLT to derive a (Gaussian) distribution for the population of the latent compartments, conditional on the observed data. Samples from this distribution can be generated, which allow the role of stochasticity to be assessed, see appendix B.3. We emphasize that the nowcast requires sample paths that are conditional on the data, for times within the inference window. Such conditional distributions cannot be sampled by direct simulation of the model, but the functional CLT enables sampling (under the assumption of large $\Omega$).

Finally, we consider stochastic trajectories that extend beyond the inference window, which we call a *stochastic* forecast. In this case, we first use a conditional nowcast to sample the latent compartments at the end of the inference window, after which we simulate the stochastic dynamics (by Gillespie [56] or tau-leaping methods [57]). This yields trajectories of the full stochastic model, with integer-valued populations. (Contrary to the conditional nowcast, these sample paths are only conditional on data from the past. Hence they can be sampled directly, due to the Markov property.)

These processes are analogous to computations with hidden Markov models (HMMs) [58]. The latent compartments correspond to the hidden variables, which are to be estimated. Also, nowcasting corresponds to sampling from the filtered distribution of the HMM, and the stochastic forecast is an HMM method for prediction. Inference based on the functional CLT leads to a multivariate Gaussian distribution for the latent variables which provides directly the filtered distribution (at this level of approximation).

# 5. Inference validation with synthetic data

To validate the methodology described so far, we consider a simple example model of SEIR type, with an additional compartment ($D$) for deceased individuals. There is a single age cohort, with population $\Omega$. The compartment populations are denoted by $S$ (susceptible), $E$ (exposed), $I$ (infectious), $R$ (recovered) and $D$ (deceased). The rates for the stochastic population model are

$$\left.\begin{array}{l} w_{S \to E} = \frac{c\beta SI}{\Omega}, \\ w_{E \to I} = \gamma_E E, \\ w_{I \to R} = \gamma_I (1-f)I \\ w_{I \to D} = \gamma_I f I. \end{array}\right\} \tag{5.1}$$

and

Here, $f$ is the infection fatality ratio (IFR), $c$ is the rate of contacts, $\beta$ is the infection probability per contact and $\gamma_E$, $\gamma_I$ are rates for progression from $E$ and $I$, respectively. Note that $\boldsymbol{n} = (S, E, I, R, D)$ is a vector of integer-valued populations, and recall from (2.3) that the corresponding fractions of the total population are $\boldsymbol{x} = \boldsymbol{n}/\Omega$. Hence (5.1) is consistent with (2.4), the rates $w$ correspond to the numbers of individuals that are transferred (on average) between the compartments, per unit time. We take $(\beta, f) = (0.035, 0.02)$ and rates $(c, \gamma_E, \gamma_I) = (20, 0.35, 0.25)$ per day.

For inference, we generate synthetic data by direct simulation of the stochastic model using Gillespie [56] or tau-leaping methods [57], depending on the population (see below). The simulation runs over an 80-day period which spans the course of the epidemic, the initial condition has $10^{-3}$ of the population in the $E$ compartment, and $4 \times 10^{-4}$ in the $I$ compartment, with all other individuals being susceptible.

We take the daily numbers of deaths as observed data from this synthetic trajectory and we attempt to infer the 'true model' (the model that generated the data). We perform inference using data from a time window that starts when the total number of deaths first exceeds 0.2% of the total population (this is a random time which depends on the stochastic trajectory). We use the methods described above to infer the rate $\beta$ and also initial conditions for the compartments $S, E, I$, denoted $S_0, E_0, I_0$. (Note, these are the initial conditions at the beginning of the inference window, not the initial conditions at time zero.) The total population $\Omega$ and the values of $(c, \gamma_E, \gamma_I)$ are fixed at their true values. (Since the initial population $D_0$ of the $D$ compartment is observed, the initial condition for $R$ is computed as $R_0 = \Omega - (S_0 + E_0 + I_0 + D_0)$.) Hence, we infer four parameters ($\beta, S_0, E_0, I_0$).

For the analysis of this section, the prior for $\beta$ is a Gaussian whose mean is 0.8 of the true value, with standard deviation one half of its mean. The prior means for $S_0, E_0, I_0$ are obtained by considering the fastest growing linear mode of the deterministic dynamics (see appendix C).

The approximate likelihood of (3.7) is accurate for large populations. As initial validation we take $\Omega = 10^8$. A trajectory of the true model is generated by tau-leaping method (stochastic generation of a full trajectory by the Gillespie method would already take a significant computational effort, comparable with the total time taken for inference of MAP parameters). We consider observed data from a 20-day inference window. We maximize the posterior over the four inferred parameters following §4.1, and we sample the full posterior distribution by MCMC following §4.2. For MCMC sampling of this model, we use an ensemble of eight walkers [54] (twice the number of inferred parameters); we take several thousand MCMC iterations per walker, resulting in a sampling time more than 50 times larger than the autocorrelation time of the underlying Markov chain. We discard one-third of the samples for burn-in of the chain.

Results are shown in figure 1. The inference machinery accurately infers all four parameters from just one stochastic trajectory. The posterior uncertainty is low—this is expected because the population of the model is very large so the CLT is an accurate description of its dynamics, and the fluctuations between trajectories are very small. (In particular, the number of deaths on each day is more than $10^4$, and the populations of $S, E, I$ classes are more than $10^6$, so the natural scale for fluctuations in these numbers is $N^{-1/2} \sim 0.1 - 1\%$.)

To illustrate model forecasting, we consider a similar situation, but now with population $\Omega = 10^6$. In this situation, daily deaths are in the 100s, so one may expect significant day-to-day fluctuations as well as some deviations from CLT behaviour. We perform inference using data from increasingly long time windows with lengths of 4–20 days; for each dataset we run independent computations of the MAP, and MCMC sampling. Figure 2 shows stochastic forecasts, as described in §4.4, as well as posterior distributions of $\beta$. As the data used for inference increases, the posterior uncertainty is reduced, as does the forecast uncertainty. Each forecast includes 40 trajectories, so the range of outcomes in each forecast can be used as a rough estimate of a 97.5% credible interval. One sees that the synthetic data

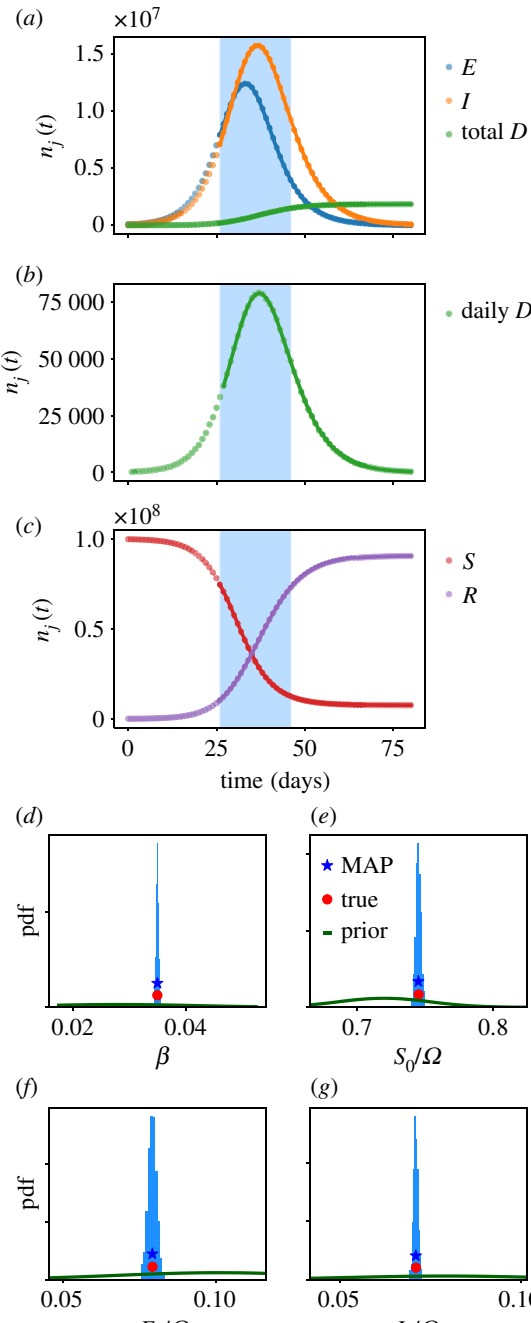

**Figure 1.** (a–c) Synthetic stochastic trajectory (points) and inferred MAP trajectory (solid lines) for the model of §5, with population $\Omega = 10^8$. The time window used for inference is shaded in blue; the daily death data from inside this window are used for inference. (d–g) Posterior histograms showing marginal parameter distributions after MCMC sampling. The true values are shown, as are the MAP estimates, and the priors. (The horizontal axes are chosen to show clearly the posterior distributions, the priors extend beyond the plotted range.) All parameters are identified accurately.

fall inside the forecast uncertainty for all time windows considered. This shows that the forecast uncertainty of the model is a reliable guideline for future behaviour.

The method validation of this section aims to establish two things. First, that the numerical implementation is adequate; and second that the approximate likelihood (3.7) yields reliable results for inference and forecasting. In this particular example, the values of the observed data are in the 100s, and we verify that the approximate likelihood (3.7) yields reliable forecasts and posterior uncertainties. Since the CLT is valid when compartment populations are large, an important question is how the performance of this methodology behaves as one considers smaller populations, especially

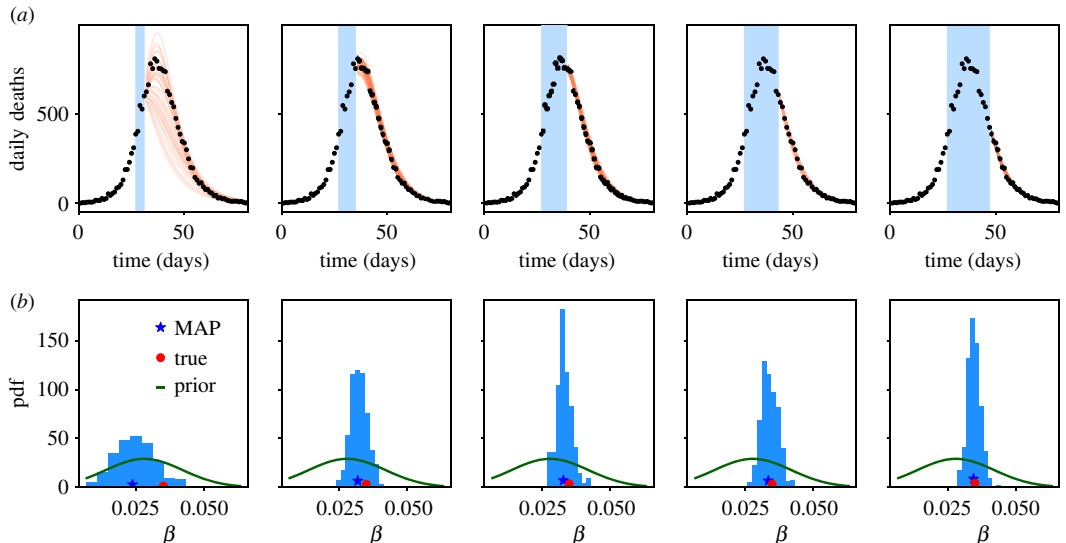

**Figure 2.** (*a*) Stochastic forecasts for the model of §5 with population $10^6$, as the amount of data used for inference is increased. Black points indicate daily deaths for a synthetic stochastic trajectory. Orange lines show 40 trajectories obtained as stochastic forecasts, based on inference using data from the blue shaded region. The forecasts converge towards the stochastic trajectory as the data used is increased. (*b*) Corresponding posterior histograms for the parameter $\beta$. The posterior uncertainty reduces as the data used increases. The longest time window is the same as that used in figure 1: the larger population in that example leads to sharper posterior estimates.

in models with more compartments. This question is a subtle one: some results are shown in appendix C, with a discussion.

As a general point, it is important that our proposed applications are for inference and forecasting based on single stochastic trajectories, as observed in epidemics. We expect in general—and the example of appendix C confirms—that parameter inference is challenging for small populations. In particular, for models with small compartment populations, the CLT approximate likelihood (3.7) will break down, which contributes to biased parameter estimates. However, we also expect large posterior uncertainties in such cases. In this situation, the results of appendix C indicate that the true model parameters are well inside the inferred posterior uncertainty, as they should be.

Finally, we remark that while inference of model parameters from synthetic data is a useful exercise, well-mixed models of real epidemics are abstractions that make strong assumptions about the disease (and surveillance) dynamics (recall §2.2). In this context, there is no 'true model'—the success of inference cannot be judged by its accuracy, but rather by its ability to fit (and forecast) the behaviour of observed time series in a consistent way, similar to figures 1 and 2. To assess this, we now apply a similar methodology to a model for COVID-19 in England and Wales.

# 6. COVID-19 in England and Wales: Model

We analyse a well-mixed compartment model for England and Wales, using data published by the Office for National Statistics (ONS), for numbers of deaths where COVID-19 was mentioned on the death certificate [59]. We consider the period 6 March to 15 May 2020, which covers the imposition of lockdown, and the associated peak in weekly deaths. (The first recorded deaths took place in the week ending 6 March, the lockdown was imposed on 23 March, and the peak in deaths was in late March and early April.) In numerical data, time is measured in weeks, starting from 6 March.

The model uses time-dependent contact structures to model non-pharmaceutical interventions (NPIs), which include the lockdown as well as other behavioural changes (mask wearing, additional hand washing, etc). For consistency with the well-mixed assumption of the model, our data excludes deaths taking place in care homes, since these individuals probably have unusual contacts, which are primarily inside their own institutions.

More precisely, individuals in the model are defined to exclude care-home residents, and we assume negligible transmission of infection from care homes to non-residents. (Note, there is no such assumption on transmission in the opposite direction, from non-residents into care homes.) We also assume (i) that

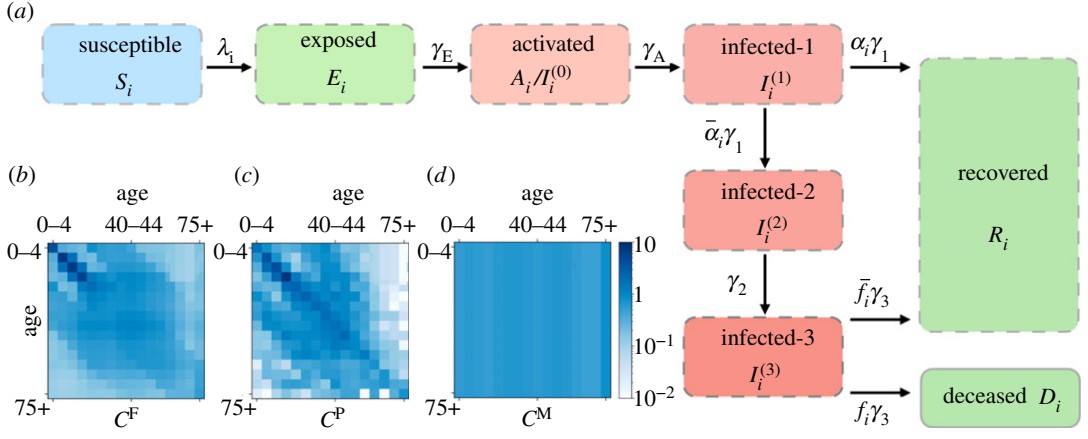

**Figure 3.** (*a*) Epidemiological classes of the example model, with rate constants indicated, the infection rate $\lambda_i$ is given in (6.2). Here, $\bar{\alpha}_i = 1 - \alpha_i$ and similarly $\bar{f}_i = 1 - f_i$. The colouring distinguishes susceptible, infectious and non-infectious compartments. The transition $I^{(1)} \rightarrow R$ represents rapid recovery of asymptomatic/paucisymptomatic cases, see main text. (*b–d*) Contact matrices for the different model variants. The colour indicates the rate of contacts between individuals in different age-groups. Specifically, each row corresponds to an age cohort for susceptible individuals who make contacts with infected individuals of various ages (corresponding to the different columns). See §6.2 for further details.

all deaths in care homes were for individuals aged 75+, and (ii) that the care-home population is small in comparison to the total, so that the $N_i$ are fixed at the total cohort populations, without being adjusted to exclude care-home residents. These assumptions simplify the model; they are not perfectly accurate, but we argue that the associated approximations are negligible compared with the (coarse) well-mixed assumption discussed in §2.2, and the uncertainties in the identification of COVID-related deaths.

Before embarking on the details of the model, we point out that it includes 128 compartments and we will infer either 46 or 47 parameters, depending on the variant. This is a challenging numerical task. It is likely that fits of similar quality could be achieved by a model with significantly fewer parameters; the dependence of parameters on age is also not very strong, so the number of age cohorts might also be reduced without much loss of accuracy. However, one purpose of this example is to test the capacity of the approach to handle models of this complexity, with a view to future work with (for example) compartments for quarantined/vaccinated individuals [40] and/or multiple variants of the virus. In this example, the inference computations are within the capability of desktop workstations, although long runs were required for MCMC and evidence computations, see below for details.

## 6.1. Definition and epidemiological parameters

We consider $M = 16$ age cohorts, which correspond to 5-year age bands from 0–4 to 70–74, and a single cohort for all individuals of age 75+. The population of cohort $i$ is $N_i$ and $\Omega = \sum_i N_i$. Given the short time period considered here, we neglect vital dynamics (birth, ageing and death by causes other than COVID-19).

The disease model is broadly consistent with other studies such as [8,10–12], although the treatment of individuals in the later stages of (more severe) disease is different, as discussed below. There are $L = 8$ epidemiological classes, illustrated in figure 3. Susceptible individuals ($S$) move to the exposed class $E$ when they become infected. The exposed class represents the latent period so these individuals are not infectious; they progress with rate $\gamma_E$ to an activated class $A$, which is infectious but non-symptomatic. We sometimes also denote this class by $I^{(0)}$. From $A$, all individuals progress to class $I^{(1)}$, with rate $\gamma_A$. Hence $I^{(1)}$ includes cases that never develop symptoms, as well as paucisymptomatic and severe cases. (Paucisymptomatic cases are defined as those with very mild symptoms, following [60].) These situations are distinguished by their progression from stage $I^{(1)}$—the total progression rate is $\gamma_1$, with an age-dependent fraction $\alpha_i$ of individuals (asymptomatic/paucisymptomatic cases) recovering into class $R$; the remainder progress to a symptomatic infectious stage $I^{(2)}$. There is progression from $I^{(2)}$ to $I^{(3)}$ with rate $\gamma_2$. After this, the (total) progression rate from $I^{(3)}$ is $\gamma_3$, of which an age-dependent fraction $f_i$ of individuals die (transition to $D$) while the remainder recover to $R$. Hence $f_i$ corresponds to the case fatality ratio (CFR), and the IFR is $(1 - \alpha_i)f_i$.

Individuals in $R$ are immune, we assume no reinfection within the period considered in this work. The inclusion of several infectious stages allows flexibility in the model as to the distribution of times between infection and recovery or death.

The infection process for cohort $i$ depends on a contact rate matrix $\tilde{C}$, the susceptibility to infection of that cohort $\beta_i$, and on how infectious is the infected individual (based on its infectious stage). Specifically, the rate for infection of individuals in cohort $i$ by those in infectious stage $k$ is

$$w_\xi(t, \boldsymbol{\theta}, \boldsymbol{n}) = \beta_i S_i \sum_{j=1}^{M} \tilde{C}_{ij}(t) \frac{v_k I_j^{(k)}}{N_j}, \tag{6.1}$$

where $S_i$ is the population of the relevant susceptible compartment, also $I_j^{(k)}(t)$ is the population of the infectious stage for cohort $j$, and $v_k$ is the infectiousness of stage $k$. There are separate transitions $\xi$ for infection of every cohort $i$, and for every infectious stage $k$. Comparing (4.2) and (6.1) shows that $K_\xi(t, \boldsymbol{\theta}) = \beta_i \tilde{C}_{ij}(t) v_k \Omega / N_j$ for this transition. The choice of contact (rate) matrix is discussed in §6.2, below. The (deterministic) equations that describe the average evolution of this model are given in appendix D.1; the force of infection for individuals in cohort $i$ (the infection rate per susceptible individual) is denoted by $\lambda_i$ and can be deduced from (6.1),

$$\lambda_i(t) = \beta_i \sum_{j} \tilde{C}_{ij}(t) \sum_{k=0}^{3} \frac{v_k I_j^{(k)}}{N_j}. \tag{6.2}$$

(Recall that $I_i^{(0)}$ should be identified as $A_i$ and $N_j$ is the total population of cohort $j$.)

Since we only consider data for numbers of deaths, it is not possible to infer all epidemiological parameters. For example, the data do not provide information about absolute numbers of cases, nor on the relative numbers of symptomatic and asymptomatic cases. For this reason, we fix the $\alpha$ and $f$ parameters to estimated (age-dependent) values based on surveillance data from Italy in the early stages of the pandemic [60]. These estimates are discussed in appendix D.2; they are subject to considerable uncertainty, but the resulting model is still flexible enough to fit the data. All remaining parameters are inferred. The $\beta$ parameters are age-dependent, all other epidemiological parameters are assumed independent of age. As noted above, the initial condition $x(0)$ must be determined from the inference parameters $\theta$. Details of this procedure and full specification of all prior distributions are given in appendix D.2.

Compared with other models such as those of [8,10–12], the main difference in our approach is that individuals in the later stages of the disease ($I^{(2)}$ and $I^{(3)}$) can still pass on the infection, albeit with reduced probabilities given by $v_2$, $v_3$ in (6.2). Such individuals have high viral load but low levels of (viable) virus in the respiratory tract [61,62], indicating $v_2 = v_3 = 0$ might be the most realistic choice as in [8,10–12]. Still the model considered here is suitable for illustrative purposes (in practice, $v_2$, $v_3 \approx 0.1$ are small, see also figure 11 below, and the associated discussion).

## 6.2. Model variants (contact matrices and NPIs)

We consider a Bayesian model comparison, based on several variants of the model described above, which differ in their contact structure.

In the absence of any NPI, infective contacts are described by (bare) contact matrices $C$, such that $C_{ij}$ is the mean number of contacts per day with individuals in cohort $j$, for an individual in cohort $i$. To account for NPIs we assume that individuals in cohort $i$ have their activities multiplied by a time-dependent factor $a_i(t) \leq 1$, so that the mean number of contacts per day during the NPI is changed to $a_i(t) C_{ij} a_j(t)$. In the absence of any intervention then $a_i = 1$. Note also, $C_{ij}$ is a number of contacts, but the quantity $\tilde{C}_{ij}$ that appears in (6.1) is a contact rate; hence we take

$$\tilde{C}_{ij} = \eta a_i(t) C_{ij} a_j(t), \tag{6.3}$$

where $\eta$ is a basic rate of $1 \, \text{day}^{-1}$. Our numerical implementation measures time in weeks, so $\eta = 7 \, \text{week}^{-1}$.

We consider three possibilities for the bare contact matrix $C$, see figure 3 and appendix D.3. Two of the choices are the matrices proposed by Prem *et al.* [63] and Fumanelli *et al.* [64], which are both based on the POLYMOD study [65]. The third is a simple proportional mixing assumption, which is that

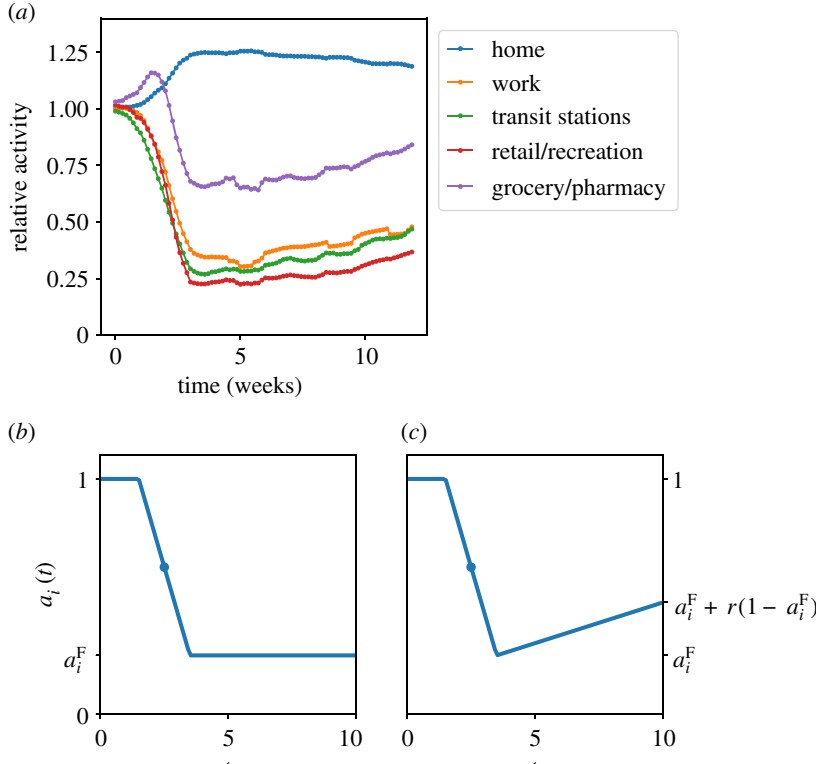

**Figure 4.** (a) Data for time spent in different activities, published by Google (UK data) [66], smoothed with a 7-day rolling average. (b) Time-dependence of step-like-NPI, (c) NPI-with-easing, the easing factor is $r$.

individuals meet each other at random

$$C_{ij} = c_0 \frac{N_j}{\Omega}, \tag{6.4}$$

where $c_0$ is a constant chosen to have a total number of contacts comparable to that of [63]. We refer to the resulting contact matrices (and the associated model variants) as $C^F$, $C^P$, $C^M$ for the models of Fumanelli *et al.* [64], Prem *et al.* [63], and proportional mixing, respectively. We do not distinguish at all between different types of contact (for example, home, work, school), the reasons for this are discussed in §7.1, below.

We also consider two possibilities for the NPI parameters $a_i(t)$, as shown in figure 4. These were chosen to mimic the patterns of activity in the UK, based on data published by Google [66]. The first possibility is a step-like-NPI, with a linear decrease from $a_i = 1$ to $a_i(t) = a_i^F$ over a time period $W_{lock}$, after which $a_i(t)$ remains constant at $a_i^F$. The mid-point of the step-like decrease is at time $t_{lock}$, the parameters of the NPI are $t_{lock}$, $W_{lock}$ and the various $a_i^F$. The second possibility is an NPI-with-easing, it involves the same step-like decrease, followed by a linear increase, such that the value at the end of the period considered is $a_i^F + r(1 - a_i^F)$, where $r$ is an additional lockdown-easing parameter (larger values correspond to more contacts). We emphasize that the Google data informed the functional forms chosen for $a_i(t)$, but all numerical parameters in this function are inferred. Priors and further model details are given in appendix D.2.

# 7. COVID-19 in England and Wales: results

We have applied the methodology of §4 to the models of §6. The total number of inference parameters (for initial conditions, epidemiological parameters and contact structure) is either 46 or 47, depending on the NPI. This number could be reduced by considering a smaller number of age cohorts, but we retain them here to illustrate that the methodology is applicable in models of this complexity.

As a baseline, we perform inference using data for the seven week period 6 March to 24 April 2020, with the remaining three weeks of our data period used to assess the resulting Bayesian forecast. For this model, converged estimates of MAP parameters are available within a few minutes on a desktop computer. For posterior sampling, we use the emcee package [54] with a number of walkers equal to twice the number of inferred variables. The estimated autocorrelation times of the underlying Markov chains were in the range 3000–5000 and sampling runs were in the range $3 \times 10^4$ to $10^5$ to ensure

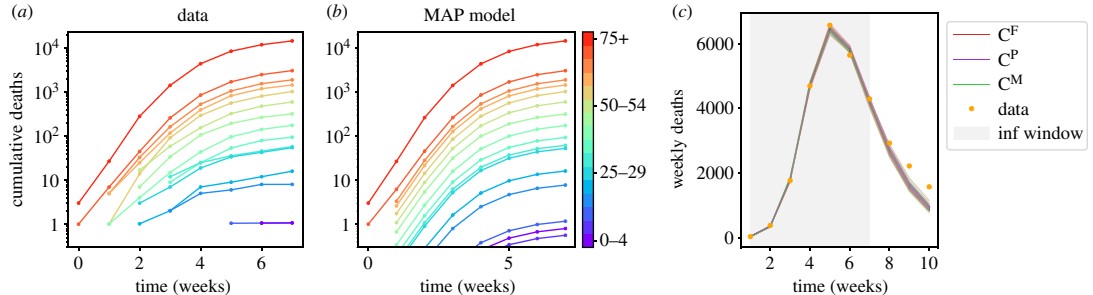

**Figure 5.** ($a$,$b$) Comparison of data with MAP trajectory for cumulative deaths ($C^F$ model variant with step-like-NPI). ($c$) Deterministic forecast for step-like-NPI with various model variants. There are 20 trajectories for each model variant, with parameters sampled from the posterior.

convergence, with the initial one-third of samples discarded to allow for burn-in. Each sampling run took several days on a single desktop workstation.

## 7.1. Step-like-NPI

Figure 5$a$,$b$ shows results for the $C^F$ model variant with step-like-NPI, and seven weeks used for inference. We show the cumulative number of deaths by cohort, for the deterministic trajectory $\bar{x}(t)$, obtained using the MAP parameter values. The model matches well the data. Note the model results are averages so the cohort populations are not integer-valued in general. Small populations (and particularly those below 1) indicate that the assumptions of the CLT are questionable, but in practice the likelihood is dominated by compartments with large populations, in which case (3.7) is still a reasonable approximation. (The data have no deaths in the 5–9 cohort, for this time period.)

Figure 5$c$ shows deterministic forecasts with step-like-NPIs (recall §4.4), based on the different contact matrices. Parameters are sampled from the posterior (as obtained by MCMC). The model variants behave almost identically and fit the data used for inference. However, the forecasts are not accurate. We attribute this primarily to lockdown easing—this is neglected within the model shown (which has $r = 0$), so an accurate forecast should not be expected. Forecasting is explored further in §7.3, including more realistic models with $r > 0$.

Figure 6$a$ shows inferred values of latent (unobserved) compartments, using a deterministic nowcast with parameters from the posterior. As expected, they show a rise and fall in the number of infected individuals, with different stages having their peaks at different times. An important set of (age-dependent) parameters are the $\beta_i$, which determine the susceptibility to infection. Figure 6$b$ shows inferred values of $\beta_i$ for the $C^F$ model, including the range of posterior samples, and the posterior mean, which are compared with the MAP estimate and the prior. The inferred values of $\beta$ are quite far from the prior mean; these parameters are very uncertain *a priori*. (This uncertainty is incorporated by using lognormal priors for the $\beta_i$ with a standard deviation one half of the mean, see appendix D.2.) The main feature in the inferred result is the large value of $\beta_i$ for the oldest cohort (75+). The inferred values of other parameters are discussed in appendix D.4; they are generally consistent with the prior assumptions.

To rationalize the inferred $\beta$, it is easily verified that for a model with the assumed contact structure, CFR and $\alpha$, the inferred value of $\beta$ for the oldest cohort must be larger than all other cohorts, in order to capture the age-dependence of deaths in England and Wales, which are very skewed towards the older age groups. There are at least two reasons why inference might lead to such a large $\beta$: either the assumed CFR (or $\alpha$) has too weak an age-dependence which is being compensated by an age-dependent $\beta$; or the contacts of elderly individuals are indeed more likely to result in infection, perhaps for medical reasons, or because of increased time in high-risk environments (such as hospitals). This distinction could be settled if accurate data for numbers of infections were included in the analysis, but it is not possible with the data considered here. The results of [11] suggest that susceptibility is age-dependent, but the dependence is weaker than we infer, indicating that both effects are in play.

Figure 6$c$ shows the (MAP) inferred $\beta$ parameters for the models with different contact matrices. While the trend is similar, there are significant differences. Nevertheless, the behaviour of the inferred models is almost identical, recall figure 5$c$. The reason is that the behaviour of the model is dominated by the infection rates of (6.1)—different contact matrices can still lead to similar model behaviour, because of the freedom to adjust the $\beta_i$. In this sense, our results can be interpreted as inference of an

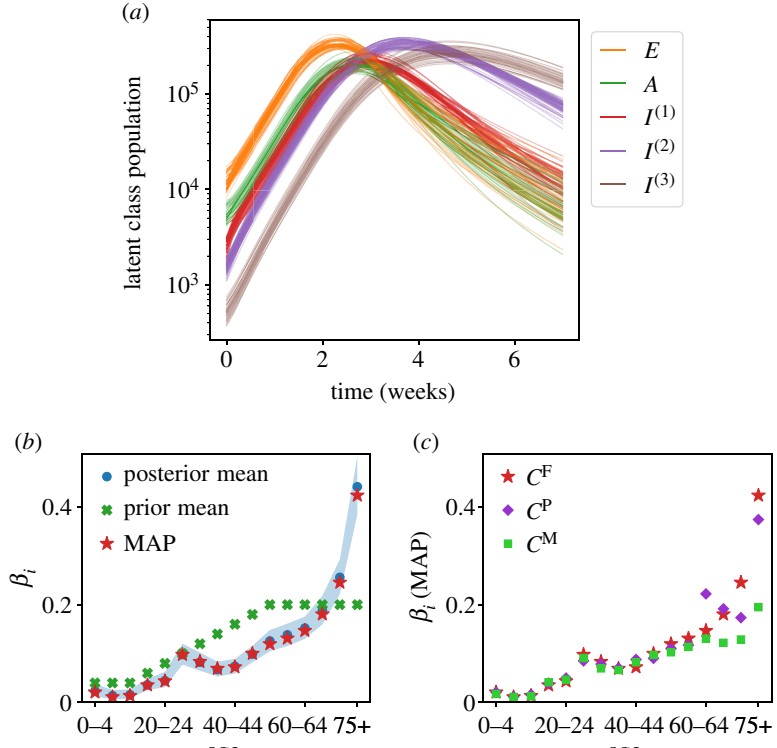

**Figure 6.** Results with step-like-NPI. (*a*) Populations of latent compartments (summed over age cohorts); 40 deterministic trajectories, corresponding to parameter samples from the posterior. (*b*) Inferred (posterior) $\beta_i$ from MCMC (using $C^F$ variant); shading shows 5th to 95th percentiles. The MAP and prior mean are also indicated. (*c*) Inferred (MAP) $\beta_i$ for different model variants.

'infective contact matrix' whose elements are $\beta_i C_{ij}$. It is notable from figure 3 that the $C^P$ contact matrix includes some large differences between cohorts with similar ages, particularly in contacts with the 75+ cohort. These can be traced back to the finite dataset of the original POLYMOD study [65]. For the $C^P$ model variant, these large fluctuations lead to an inferred $\beta_i$ with a complicated dependence on age, for cohorts in the 60+ group. In the $C^F$ variant, the dependence on age is much smoother, both for contacts and for $\beta$. Compared with the contact matrices that are based on POLYMOD [63,64], the $C^M$ variant has (much) more contacts for older individuals, so the inferred $\beta$ is lower in the older cohorts.

## 7.2. Fisher information matrix and model evidence

We now discuss the FIM (4.3) for the $C^F$ model variant with step-like-NPI. Two items of particular interest are parameters $\theta_a$ whose inferred values are very sensitive to the data, and soft modes of the parameter space along which the likelihood varies slowly. These modes indicate aspects of the model that are mostly determined by the prior.

The sensitivities of (4.4) provide useful information on the first point. Figure 7 shows the results. The parameters most sensitive to the data are the rates $\gamma_E$, $\gamma_A$ and $\gamma_1$, the probability of the oldest age cohort to get infected $\beta_{75+}$, and the time of lockdown $t_{lock}$, consistent with the discussion so far. These parameters have $s_a > 100$, indicating that changes of order 1% in their values are sufficient to change the log-likelihood by an amount of order unity.

Soft directions around the MAP parameters, in which the model behaviour is expected to change very little, do exist. They arise from small eigenvalues of the FIM, and the corresponding eigenvectors. One example of such a soft mode is discussed in appendix D.4. The existence of soft modes speaks in favour of a Bayesian approach, in that prior information about the disease is used to fix those parameters which are not determined by the data. This makes best use of all information sources, including expert-derived priors.

We have also computed the evidence for these models, see figure 8. The $C^F$ and $C^P$ contact matrices lead to similar log-evidences, with the $C^P$ variant higher by around 3 units (we use natural logarithms throughout). The contact matrix with proportional mixing leads to log-evidence that is smaller by around

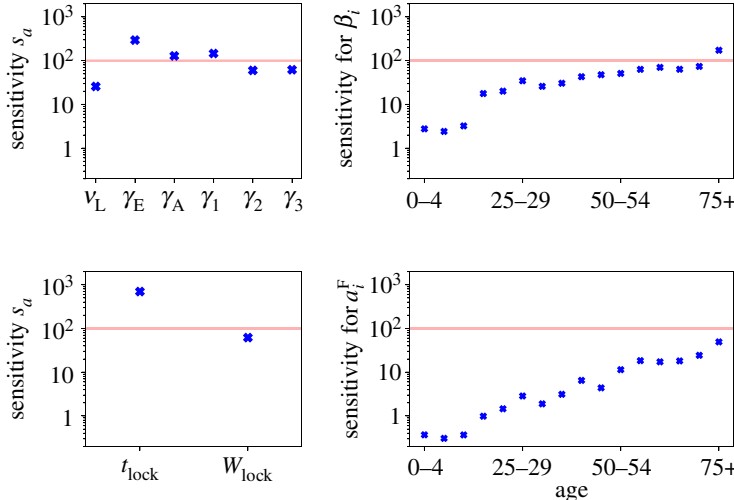

**Figure 7.** Fisher information matrix: sensitivities for model parameters. Red lines show the value 100, as an (arbitrary) indication of parameters that are very sensitive to the data. Similar results for the parameters that determine the initial condition are given in appendix D.4.

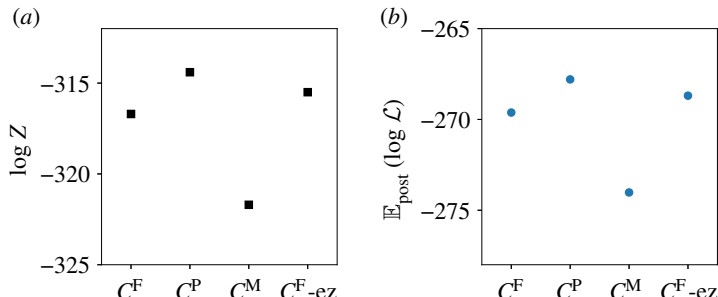

**Figure 8.** (a) Model evidence for different model variants. The notation $C^F$-ez indicates the $C^F$ model variant and NPI-with-easing; all other results are for step-like-NPI. (b) Posterior mean log-likelihood: this indicates the extent to which the inferred models fit the data.

8 units. We conclude that this model can still fit the data with reasonable accuracy, but the inference computation is sensitive enough to infer that the contact structure has some assortativity. Also shown is the posterior mean of the log-likelihood $\mathbb{E}_{post}[\log \mathcal{L}]$ which is the negative of the deviance, recall (4.7). This similar behaviour of the evidence and deviance indicates that the differences between the models are primarily in the quality of the fit, rather than the amount of fine-tuning required for the parameters.

Given the very naive assumptions of the proportional mixing model, we argue that the difference of 8 units in log-evidence should be regarded as a mild effect. Our conclusion is that the inference computation is not extremely sensitive to prior assumptions on the contact structure. Based on this result, it seems that more detailed modelling of contacts (for example, separation by work/home/ school) will have relatively little impact on the quality of inference, given the very large uncertainties within the model about the values of $\beta_i$.

## 7.3. NPI-with-easing

We now consider NPI-with-easing. Since the behaviour with different contact matrices is very similar, we restrict to the $C^F$ model variant.

Figure 9a is a deterministic forecast analogous to figure 5c; it shows how the easing parameter $r$ leads to increased uncertainty in the forecast, in a way that is more consistent with the data. By contrast, figure 9b shows a stochastic forecast as defined in §4.4. This accounts for stochasticity in the epidemiological dynamics, it automatically matches the data within the inference window. The results of the two kinds of forecast are similar, indicating that the dominant source of uncertainty is coming from the model parameters.

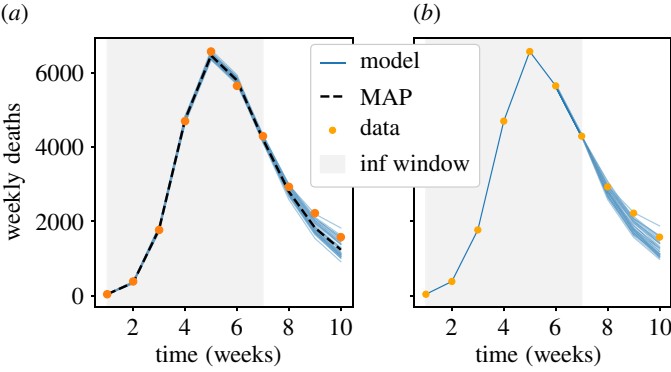

**Figure 9.** Deterministic and stochastic forecasts, NPI-with-easing, $C^F$ model variant. (*a*) Deterministic (averaged) forecast, 40 trajectories with parameters from posterior; (*b*) stochastic forecast conditional on data. Compared with figure 5*c*, the effect of lockdown easing is to increase deaths at later times, which improves the agreement with data.

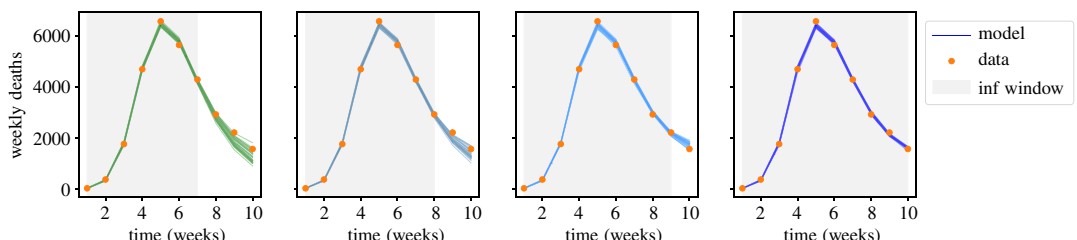

**Figure 10.** Deterministic forecasts showing the effect of increasing the time period used for inference from 7 to 10 weeks. ($C^F$ model variant; NPI-with-easing.)

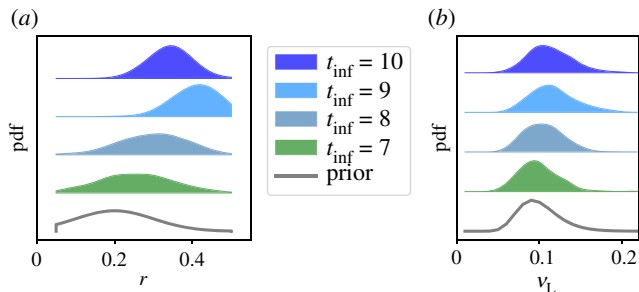

**Figure 11.** Posterior distributions of the easing parameter $r$, and the late-stage infectiousness factor $v_L$, as the inference time period $t_{inf}$ is increased. ($C^F$ model variant; pdfs are shown as Gaussian kernel density estimates.)

To explore the effects of lockdown easing in more detail, we consider the effect of increasing inference window, always comparing the model forecasts for the same 10-week period. Results are shown in figure 10. The agreement between inferred model and the data increases, as expected—we find that this model can accurately fit the data, with reasonable parameter values.

For the seven-week inference window, figure 11 shows that the distribution of $r$ is still close to the prior. This is consistent with the result of figure 8, that the evidence of the variant with easing is comparable to the variants with step-like-NPI. That is, the additional parameter $r$ leads to a mild improvement in the fit to the data, and fine-tuning of its value is not required.

When considering longer time windows, we note that deaths are lagging indicator of the number of cases, which means that $r$ is still not fully determined by the data. That is, these results still depend significantly on the prior (for details see appendix D.2). Nevertheless, increasing the inference window causes the posterior distribution of $r$ to shift towards larger values, leading to improved agreement with the data. There are also significant differences in these posterior distributions, for example if 9 or 10 weeks data are used for inference—this limits the robustness of the forecasting and indicates a possible

tendency to overfitting. We attribute this primarily to the simple linear easing assumed in our NPI. Most other parameters depend weakly on the period used for inference (see appendix D.4).

The posterior distributions for $v_L$ in figure 11 are also similar to the prior, showing that this parameter is weakly identifiable. As noted above, an alternative modelling hypothesis would be that $v_L = 0$, as in [8,10–12], consistent with the results summarized in [62]. In this work, that possibility is suppressed by the (lognormal) prior for $v_L$. The expert judgement of [62] might be used to refine the model by adjusting this prior—this illustrates the adaptability of the Bayesian framework.

In evaluating these results, we note that both the model and the likelihood assume a well-mixed population. In practice, individuals have correlated behaviour, which can be expected to enhance stochastic fluctuations. For this reason, it is likely that the functional CLT underestimates the variance of the data, given the model. This can lead to an overfitting effect. There are also uncertainties in the data that are not accounted for in the likelihood, such as possible under-detection of COVID-related deaths in the early period of the epidemic. Recalling that the deceased population in the model includes only those individuals who were diagnosed with COVID-19, such an under-detection might be modelled (in this framework) by a time-dependent CFR.

# 8. Discussion

We have described a methodology for inference and forecasting in epidemiological compartment models, where all stochastic aspects of disease propagation and measurement are modelled on an equal footing, and the likelihood is justified from first principles and derived directly from the model. This means that the likelihood can be computed directly from the model definition, given appropriate data.

## 8.1. Example model

This methodology has been used to calibrate a model for the COVID-19 pandemic in England and Wales, based on death data. We have compared models with different contact structures, showing that fine details of the contact matrix have very little effect on model behaviour and forecasts. Indeed, the model with proportional mixing behaves very similarly to those with contact matrices derived from the POLYMOD dataset [63–65]. This may be surprising at first glance, but the fact that the $\beta_i$ parameters are inferred separately for each cohort means that the model has enough flexibility to infer how many infective contacts are made by each group. More specifically, it is the infection rate constant $K$ of (4.2) that determines the model behaviour, so one sees from (6.1) differences in contact matrices can be partially compensated by changes in $\beta$. (The compensation is only partial because while $\beta_i$ controls the relative numbers of infections, the contact matrix also determines the assortativity of mixing.)

In contrast to the details of mixing among cohorts, modelling assumptions about time-dependence of the contact structure have a significant impact on forecasting, as one should expect. This is illustrated by the dependence of the behaviour on the easing factor $r$.

Within the time period considered, the model gives forecasts that are reasonably accurate and robust. However, we have identified a possible tendency to overfitting, some of which may be due to the well-mixing assumption that is used in the likelihood. Another common approach uses negative binomial distributions in the likelihood [10,11,18], this corresponds to a larger variance for numbers of deaths (overdispersion), compared with the CLT. It would be interesting to consider inclusion of an over-dispersion factor in the likelihood used here, as a way of accounting for correlations in the contact structure.

In terms of model calibration, the main limitation of this study is the fact that we do not use data for case numbers, which means that the CFR cannot be inferred. In the UK, the rates and policies for testing for COVID-19 have had complex time-dependence, which means that robust estimation of case numbers is challenging. Incorporation of a time-dependent testing capacity into this framework is a direction of ongoing research. The extension of this framework to geographically resolved models is also under active investigation.

## 8.2. Methodology: strengths and weaknesses

The example models of §5 and 6 show that the methodology is effective in population-level models of large epidemics. These are situations in which the CLT approximation to the likelihood is expected to

be valid, so they should fall within the applicability of these methods. We repeat that for models with small populations (where demographic noise becomes very large), models that do not rely on the CLT approximation should be preferred [37–39]. Compared with inference methods with deterministic disease dynamics [10–12], the approach is somewhat more expensive, because of the requirement to compute the CLT covariance for the trajectory. Still, the examples show that relatively complicated models are still within reach.

For the simple model of §5, accurate parameter estimation is possible when the population is very large, based on a single stochastic trajectory. For smaller populations, experiments on single trajectories show that the posterior uncertainty grows, which is again consistent with theoretical expectations. As this happens, the true model parameters remain inside the posterior credible intervals, as they should. Hence, while the posterior distributions may suffer some bias due to deviations from CLT behaviour, they are still reasonable estimates of parameter uncertainty.

For the model of England and Wales in §6, the scheme infers parameters that fit the data, and the forecast of figure 10 indicates that the posterior distributions are reasonably accurate, even when considering more than 40 parameters. Given the modelling assumptions (particularly the well-mixed assumption, without over-dispersion), this result shows that the method has promise. Further work on more detailed and accurate models [40] will provide new and stringent tests of its applicability.

Data accessibility. The example models of §5 and 6 were analysed using the PyRoss library [31]. The analysis codes and the resulting data are available at https://github.com/rljack2002/infExampleCovidEW (this includes the example with synthetic data, and the example model for England and Wales). This data will also be available at https://doi.org/10.17863/CAM.72839.

Authors' contributions. The integration of epidemiological modelling, the Bayesian estimation of models against epidemiological data, and the optimization of NPIs in these fitted models was conceived by R.A., who also led the PyRoss project, with additional guidance from M.E.C. Development of the model for England and Wales was led by R.L.J. The inference methodology was developed by Y.I.L., G.T., P.B.R. and P.P., with specific contributions including functional CLT (Y.I.L.); Fisher information matrix (G.T.); evidence and MCMC (P.B.R.); and likelihood computation (P.P.). Also, J.K. developed stochastic simulation and forecasting; R.S. designed the numerical implementation of PyRoss and developed deterministic simulations; and J.D. developed the example model for England and Wales. Contributions by T.E., L.K., A.B. and J.D.P. enabled flexible inference for compartment models, and H.K. contributed to the stochastic forecasts. The manuscript was written by R.L.J. and R.A., with contributions from all authors.

Competing interests. We declare we have no competing interests.

Funding. This work was undertaken as a contribution to the Rapid Assistance in Modelling the Pandemic (RAMP) initiative, coordinated by the Royal Society. This work was funded in part by the European Research Council under the Horizon 2020 Programme, ERC grant no. 740269, and by the Royal Society grant no. RP17002. The authors are also grateful for financial support from the EPSRC doctoral training programme (A.B., J.D., G.T.), the Leverhulme Trust (P.B.R. and H.K.), the Cambridge Trust and Jardine foundation (Y.I.L.).

Acknowledgements. We thank Graeme Ackland, Daniela de Angelis, Paul Birrell, Daan Frenkel, Sanmitra Ghosh and Ken Rice for helpful discussions. We also thank PyRoss contributors, Fernando Caballero, Bilal Chughtai, Jules Guioth and Benjamin Remez; and JPM researchers William Bankes, Erik Brorson, Andrew Ng and William Peak.

# Appendix A. Derivation of $G$ for likelihood

This appendix derives the covariance matrix $G$ that appears in the likelihood (3.7). The result is based on the CLT for $u$ discussed in §2.4. We first compute a covariance matrix $\tilde{G}$ for the state $u$, from which we derive the covariance $G$ of the data.

Note that $J$ and $\sigma_\xi$ in (2.7) depend on the deterministic path $\bar{x}$ but not on the random variable $u$, so (2.7) is a time-dependent Ornstein–Uhlenbeck process. This enables derivation of three important results. The first concerns the covariance matrix $\Sigma$ for $u(t)$, whose elements are

$$\Sigma_{ij}(t) = \langle u_i(t) u_j(t) \rangle. \tag{A 1}$$

Here and throughout, angled brackets $\langle \cdot \rangle$ denote an average over the stochastic dynamics of the compartment model. (Recall that $u(0) = 0$ so $\langle u(t) \rangle = 0$ and also $\Sigma_{ij}(0) = 0$.) The equation of motion for $\Sigma$ can be derived from (2.7) as

$$\frac{\partial}{\partial t} \Sigma(t) = J(t, \boldsymbol{\theta}, \bar{x}_t) \Sigma(t) + \Sigma(t) J^T(t, \boldsymbol{\theta}, \bar{x}_t) + B(t, \boldsymbol{\theta}, \bar{x}_t), \tag{A 2}$$

where

$$B(t, \boldsymbol{\theta}, \overline{\boldsymbol{x}}_t) = \sum_\xi \boldsymbol{\sigma}_\xi(t, \boldsymbol{\theta}, \overline{\boldsymbol{x}}_t) \boldsymbol{\sigma}_\xi^T(t, \boldsymbol{\theta}, \overline{\boldsymbol{x}}_t),$$ (A 3)

is a square matrix. Equation (A 2) can be solved (numerically) for $\Sigma$.

Second, let $\langle \cdot \rangle_{\boldsymbol{u}(s)}$ denote an average, conditional on the value of $\boldsymbol{u}(s)$. Taking the first moment of (2.7) one arrives at a linear equation for the average value of $\boldsymbol{u}$, hence for $t \geq s$ one has

$$\langle u_i(t) \rangle_{\boldsymbol{u}(s)} = \sum_j U_{ij}(s, t) u_j(s),$$ (A 4)

where the matrix $\boldsymbol{U}(s, t)$ is the time-evolution operator, which solves the linear differential equation

$$\frac{\partial}{\partial t} U_{ij}(s, t) = \sum_k J_{ik}(t, \boldsymbol{\theta}, \overline{\boldsymbol{x}}_t) U_{kj}(s, t),$$ (A 5)

with initial condition $U_{ij}(s, s) = \delta_{ij}$. This equation is readily solved (numerically) for $\boldsymbol{U}$.

The third result can then be obtained by noting that the covariance matrix for $\boldsymbol{u}$ between two times $s, t$ can be obtained (for $t \geq s$) by composing the covariance $\Sigma(s)$ with the propagator $\boldsymbol{U}(s, t)$. That is,

$$\langle u_j(s) u_i(t) \rangle = \sum_k \Sigma_{jk}(s) U_{ik}(s, t).$$ (A 6)

To exploit this last result, consider the vector obtained by concatenating the full state of the system over observed time points, analogous to (3.3)

$$X = \big(x(t_1), x(t_2), \dots \big).$$ (A 7)

Its mean is clearly $\overline{X} = (\overline{x}(t_1), \overline{x}(t_2), \dots)$. Now define a (scaled) deviation from the mean as $\tilde{\boldsymbol{\Delta}} = (X - \overline{X})\sqrt{\Omega}$, and denote the covariance of this vector by $\tilde{G}$. From (A 6), this symmetric matrix is formed of blocks that depend on $\Sigma$ and $\boldsymbol{U}$:

$$\tilde{G} = \begin{pmatrix} \Sigma(t_1) & \Sigma(t_1) \boldsymbol{U}^T(t_1, t_2) & \cdots \\ \boldsymbol{U}(t_1, t_2) \Sigma(t_1) & \Sigma(t_2) & \cdots \\ \boldsymbol{U}(t_1, t_3) \Sigma(t_1) & \boldsymbol{U}(t_2, t_3) \Sigma(t_2) & \cdots \\ \vdots & \ddots & \ddots \end{pmatrix}.$$ (A 8)

Since (2.7) is an Ornstein–Uhlenbeck process, it can be shown additionally [44–46] that the distribution of $\tilde{\boldsymbol{\Delta}}$ is asymptotically Gaussian, with the given covariance $\tilde{G}$. Since the observed data are related linearly to $X$ according to (3.2, 3.3), one then obtains (3.7), with the covariance of $\boldsymbol{\Delta}$ given by

$$G = F\tilde{G}F^T.$$ (A 9)

Since all elements of $\tilde{G}$ can be evaluated, this allows computation of the likelihood (3.7).

# Appendix B. Implementation details for inference

## B.1. Fisher information matrix

For a multivariate normal distribution, such as the likelihood obtained in §3, with the mean vector $\overline{Y}(\boldsymbol{\theta})$ and the covariance matrix $G(\boldsymbol{\theta})$ the elements of the FIM (4.3) are [67]

$$\mathcal{I}_{a,b} = \frac{\partial \overline{Y}^T}{\partial \theta_a} G^{-1} \frac{\partial \overline{Y}}{\partial \theta_b} + \frac{1}{2} \mathrm{tr} \left( G^{-1} \frac{\partial G}{\partial \theta_a} G^{-1} \frac{\partial G}{\partial \theta_b} \right).$$ (B 1)

This form is advantageous since its computation only requires first-order derivatives, which are estimated as finite differences.

## B.2. Evidence computation

We use a thermodynamic integration method [68,69] to compute the log-evidence

$$\log Z = \log \int_{\mathcal{D}} e^{\mathcal{A}(\theta)} P(\theta) \, d\theta, \tag{B 2}$$

where $\mathcal{A}(\boldsymbol{\theta}) = \log \mathcal{L}(\boldsymbol{\theta})$ is the log-likelihood, and the domain $\mathcal{D}$ is the support of the prior $P(\boldsymbol{\theta})$. Compared with alternatives (for example, nested sampling [70]), thermodynamic integration allows robust estimation of convergence, and its numerical uncertainties.

We summarize the method, which is based on an integration path from a tractable (and normalized) distribution

$$\tilde{P}(\theta) = e^{\tilde{\mathcal{A}}(\theta)} P(\theta), \tag{B 3}$$

to the posterior. To this end, define

$$f(z) = \log \int_{\mathcal{D}} e^{z[\mathcal{A}(\theta) - \tilde{\mathcal{A}}(\theta)]} \tilde{P}(\theta) \, d\theta, \tag{B 4}$$

so that $f(0) = 0$ and $f(1) = \log Z$. Differentiating yields

$$f'(z) = e^{-f(z)} \int_{\mathcal{D}} [\mathcal{A}(\theta) - \tilde{\mathcal{A}}(\theta)] \, e^{z[\mathcal{A}(\theta) - \tilde{\mathcal{A}}(\theta)]} \tilde{P}(\theta) \, d\theta. \tag{B 5}$$

The right-hand side is an expectation value $\mathbb{E}_{\pi_z}[\mathcal{A} - \tilde{\mathcal{A}}]$ with respect to the (normalized) intermediate distribution $\pi_z(\theta) = e^{-f(z) + z[\mathcal{A}(\theta) - \tilde{\mathcal{A}}(\theta)]} \tilde{P}(\theta)$. Given some $\tilde{\mathcal{A}}$, this expectation value can be estimated by MCMC. (Our choice for $\tilde{\mathcal{A}}$ is discussed just below.) Then the log-evidence can be expressed as

$$\log Z = \int_0^1 f'(z) \, dz. \tag{B 6}$$

To estimate this integral, we take a sequence $0 = z_0 < z_1 < \cdots < z_n = 1$ and we use trapezoidal quadrature

$$\log Z \approx \sum_{i=1}^{n} (z_i - z_{i-1}) \frac{f'(z_i) + f'(z_{i-1})}{2}. \tag{B 7}$$

For each quadrature point, $f'(z_i)$ is estimated by an MCMC computation of the expectation value (B5). These are independent MCMC estimates, which facilitates analysis of numerical uncertainties.

To select a suitable $\tilde{\mathcal{A}}$, the idea is that the closer is $\tilde{P}$ to the posterior, the shorter is the integration path, and the easier the computation. However, $\tilde{P}$ must be a normalized distribution on $\mathcal{D}$. A suitable choice for $\tilde{P}$ is therefore a truncated Gaussian approximation to the posterior, around the MAP parameters $\boldsymbol{\theta}^*$. Let $H$ be the Hessian matrix of the (negative) log-posterior, whose elements are

$$H_{ab} = -\frac{\partial^2}{\partial \theta_a \partial \theta_b} [\log \mathcal{L}(\boldsymbol{\theta}^*) + \log P(\boldsymbol{\theta}^*)]. \tag{B 8}$$

Then the truncated Gaussian approximation of the posterior distribution is

$$\tilde{P}(\theta | Y) \propto \exp\left( -\frac{1}{2} (\theta - \theta^*)^T H (\theta - \theta^*) \right) \tag{B 9}$$

for $\theta \in \mathcal{D}$, and $\tilde{P} = 0$ otherwise. Rejection sampling is used to sample this distribution and to simultaneously obtain its normalization constant, so that $\tilde{\mathcal{A}}(\boldsymbol{\theta}) = \log(\tilde{P}(\theta|Y)/P(\theta))$ can be computed, consistent with (B 3). Hence (B 7) can be computed.

## B.3. Conditional nowcast

To sample latent compartments during the inference period, we use the CLT for $\boldsymbol{u}$ discussed in §2.4. It is convenient to assume that the latent populations are to be inferred at the times $t_\mu$ where data was collected. (This assumption is easily relaxed, at the expense of some heavier notation.) Hence, the populations of the latent compartments are encoded in the vector $X$ of (A 7). Given the model parameters, the distribution of $X$ obeys a CLT

$$\log P(X | \boldsymbol{\theta}) \simeq -\frac{1}{2} [\boldsymbol{\Delta}^T \tilde{G}^{-1} \boldsymbol{\Delta} + \ln \det(2\pi \tilde{G})], \tag{B 10}$$

analogous to (3.7), with $\tilde{G}$ as in (A 8). Since this distribution is (multivariate) Gaussian, and the data depend linearly on $X$, it is straightforward to condition on the data and obtain a Gaussian distribution for the latent compartments, which can then be sampled.

# Appendix C. Details and additional results for simple SEIR model

## C.1. Priors (including use of linearized dynamics)

For the example of §5, the Gaussian prior for $\beta$ was described in the main text. In addition, we note that all priors are truncated to avoid negative values for parameters, as well as very large values. (This truncation has very little effect on the inferred parameters.)

Priors are also required for $S_0, E_0, I_0$, which are compartment populations at the start of the inference period. Denoting this time by $t_0$, this means that the vector $x(t_0)$ must be determined from the inference parameters $\theta$ (together with the observed value of the $D$ compartment).

A convenient estimate of $x(t_0)$ is available by linearizing the average dynamics (2.5) about the state $x_S$ where all individuals are susceptible. The behaviour of the resulting equation (at early times) is dominated by the largest eigenvalue of the matrix $J(0, \theta, x_S)$, as obtained from (2.8). The corresponding eigenvector dominates the evolution of the early stages of the epidemic, up to transient effects of the initial condition, which are controlled by the smaller (sub-dominant) eigenvalues. A suitable baseline estimate for the initial condition is then

$$x_{\mathrm{lin}} = x_S + \kappa v_{\theta}^*, \tag{C 1}$$

where $v_{\theta}^*$ is the dominant eigenvector (which depends on the epidemiological parameters), and $\kappa$ is a parameter. The normalization of the eigenvector is $\sum_{\alpha} |v_{\theta,\alpha}^*| = 1$ where $v_{\theta,\alpha}^*$ is the $\alpha$-th element of the vector $v_{\theta}^*$; this means that the value of $\kappa$ is approximately one half of the non-susceptible (infected + recovered + deceased) fraction of the population, at $t = t_0$.

In the example of §5, this linearization is used to fix the prior for the initial condition parameters $S_0, E_0, I_0$: the eigenvector $v_{\theta}^*$ is computed for the true model epidemiological parameters. The fraction of deceased individuals at $t_0$ is known, which is used to fix $\kappa$, leading to estimates for $S_0, E_0, I_0$. These estimates are used for the prior mean. The prior distributions are taken to be Gaussian: the prior standard deviations for $E_0, I_0$ are one-third of their means; the prior standard deviation for $S_0$ is equal to that of $E_0$.

## C.2. Additional results (effect of smaller populations)

Figure 12 shows results of an inference computation for a total population $\Omega = 10^4$, similar to figure 1 of the main text. One sees that the numbers of daily deaths are mostly in single digits, so the CLT approximation (3.7) is not expected to be fully accurate. Still, the MAP trajectory provides a very reasonable fit to the data. The posterior distribution of $\beta$ is significantly narrower than the prior, and both posterior mean and MAP values are close to the true value. The posterior distributions for initial conditions follow quite closely the prior, indicating that the data are not sufficient to identify their values. However, the inference machinery does find significant posterior correlations among the initial conditions, even if their marginals are broad. This shows that the data do constrain these parameters significantly.

Figure 12 also shows that the method infers significant posterior correlations between the parameters (which are independent under the prior). There is a correlation between $E_0$ and $I_0$ because the model is more sensitive to the total number of initial infections $E_0 + I_0$ than to the difference between $E_0$ and $I_0$. Also, models with similar likelihood to the true model can be obtained by assuming a significant recovered population ($R_0$) at time $t_0$, this reduces the susceptible population, which can be compensated by an increased $\beta$. The resulting models also provide reasonable fits to the data.

Finally, we consider the possibility that the approximate likelihood (3.7) might lead to biased (or misleading) estimates of parameter values when populations are small (so the CLT breaks down). A Bayesian analysis of bias in this context would consider the effect on inference of providing increasing quantities of data. However, such a situation is not realistic for epidemiological applications in which one typically has data from a single epidemic (or outbreak), to be used for inference and forecasting.

To explore this situation, we mimicked a practical application of the method, as follows. We repeated the numerical experiment of figure 12 with 16 independent sets of synthetic data. For these computations, the prior mean for $\beta$ was set to its true value, to avoid trivial bias on its posterior estimate. For each experiment, we computed the posterior mean $\beta$, and its 95% credible interval (CI) from the sampled posterior histogram.

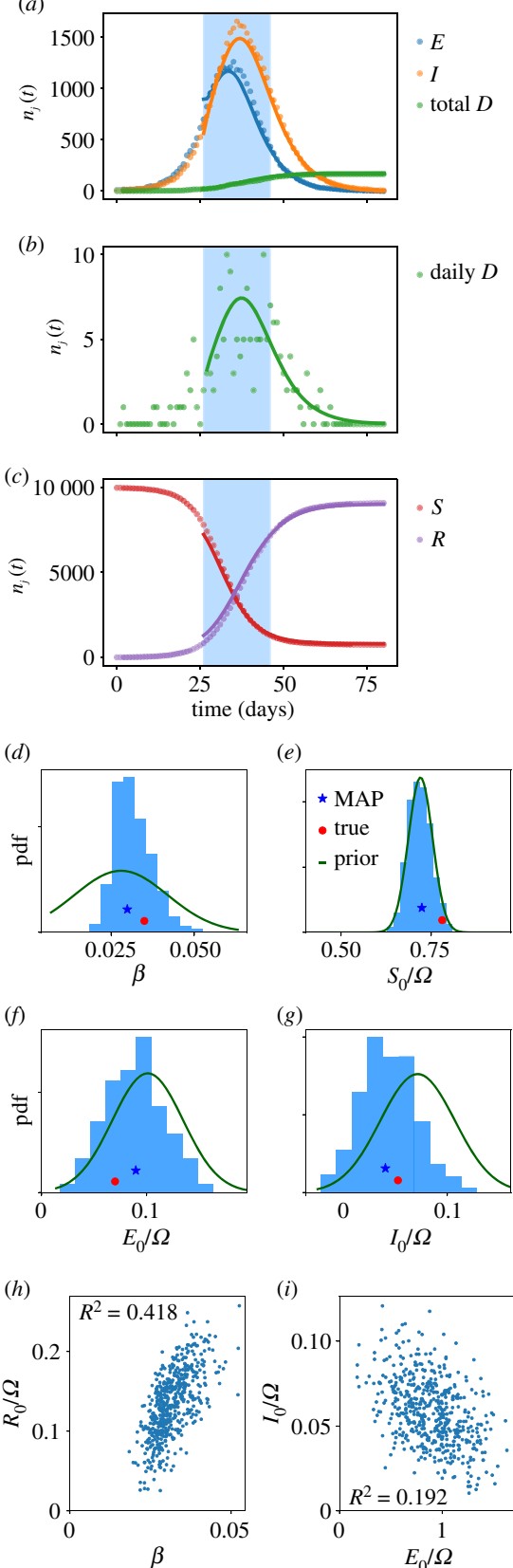

**Figure 12.** Inference based on synthetic data with relatively small population $\Omega = 10^4$: compare with figure 1. (a–c) The synthetic trajectory is shown as points, the (deterministic) MAP trajectory is shown with solid lines, showing reasonable agreement to the data. (d–g) Posterior histograms show that posterior uncertainty is reduced with respect to the prior, but it is still significant. (h,i) Scatter plots, illustrating some inferred posterior correlations, $R^2$ is a Pearson correlation coefficient.

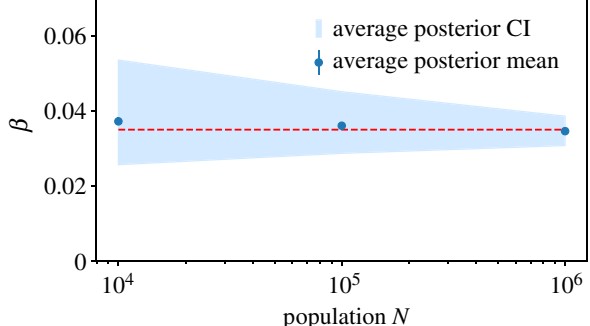

**Figure 13.** Inference at small populations in the example model of §5. The data points show the average value of the posterior mean estimate of $\beta$, obtained by repeating the inference for 16 independent synthetic datasets. Error bars (standard error on the estimate of the average posterior mean) are smaller than symbol sizes. For each dataset, we also estimate the 95% credible interval. As an indication of posterior uncertainty, we have averaged the lower and upper bounds of this CI, shown as a shaded region. The posterior uncertainty is much larger than the apparent bias. (The corresponding prior CI coincides approximately with the vertical axis.)

From these 16 experiments, we took the average of the posterior mean and the average CI, which are shown in figure 13, together with similar experiments with larger populations, up to $10^6$. The average posterior mean is close to the true value, even for small populations—this indicates that the approximate likelihood does not lead to large systematic errors in estimates of this parameter. Instead, the main effect of reducing the population is that the inferred CI on $\beta$ becomes increasingly wide. This is expected because the approximate likelihood (3.7) is proportional to $\Omega$, leading to sharp parameter estimates at large population, but large uncertainty when numbers are smaller. The same message is apparent from figure 12: reduction of the population leads to broad posterior histograms with the true value well inside the credible range.

Of course, these results do not establish that the approximate likelihood (3.7) will not lead to biased (or misleading) estimates in some situations, because of breakdown of the CLT in small populations. On the other hand, these results serve as a stress-test for the method: it does not lead to systematic errors or misleading estimates of uncertainty in this example, even when daily observations are in single digits, outside the strict range of validity of the CLT.

# Appendix D. Details and additional results for the example model of COVID-19

## D.1. ODEs for average dynamics

For the (stochastic) model defined in §6, the deterministic equations for the mean (2.5) can be written in terms of the compartment populations. For this section alone, let $S_i$ be the *average* population of susceptible individuals in cohort $i$, and similarly for all other classes. Our notation omits the dependence of these populations on time, for compactness. Using dots to indicate time derivatives, (2.5) becomes

$$
\left.\begin{aligned}
\dot{S}_i &= -\lambda_i(t)S_i \\
\dot{E}_i &= -\gamma_{\mathrm{E}}E_i + \lambda_i(t)S_i \\
\dot{A}_i &= -\gamma_{\mathrm{A}}A_i + \gamma_{\mathrm{E}}E_i \\
\dot{I}_i^{(1)} &= -\gamma_1 I_i^{(1)} + \gamma_{\mathrm{A}}A_i \\
\dot{I}_i^{(2)} &= -\gamma_2 I_i^{(2)} + \bar{\alpha}_i \gamma_1 I_i^{(1)} \\
\dot{I}_i^{(3)} &= -\gamma_3 I_i^{(3)} + \gamma_2 I_i^{(2)} \\
\dot{R}_i &= \alpha_i \gamma_1 I_i^{(1)} + \bar{f}_i \gamma_3 I_i^{(3)} \\
\dot{D}_i &= f_i \gamma_3 I_i^{(3)},
\end{aligned}\right\} \tag{D 1}
$$

and

where we do not indicate the dependence of the compartment populations on time (for compactness of notation), while $\bar{\alpha}_i = 1 - \alpha_i$ and $\bar{f}_i = 1 - f_i$, also $\lambda_i$ is given by (6.2).

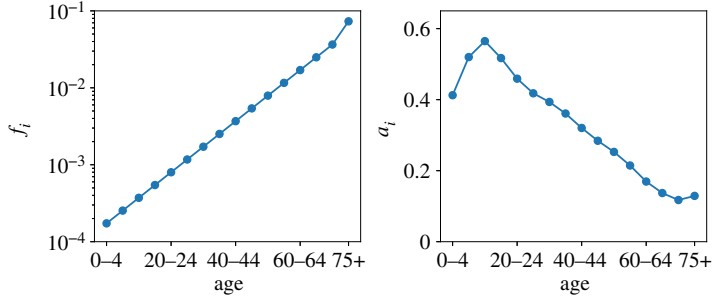

**Figure 14.** Age-dependent parameters for the CFR $f_i$ and the fraction of asymptomatic/paucisymptomatic cases $\alpha_i$.

**Table 1.** Priors for the compartment model, all parameters are independent with normal or lognormal distributions, as shown. The standard deviation (s.d.) and bounds are quoted relative to the prior mean. The prior mean for $\beta$ depends on the age cohort: $\beta_i = 0.2$ for ages 50+ and $\beta_i = 0.04$ for ages less than 15, with linear interpolation in the intermediate range. This age-dependence is based on [11], the overall scale was chosen so that the prior mean model is broadly consistent with exponential growth of cases in the first few weeks of the epidemic.

|            | distribution | mean                   | s.d./mean | bounds/mean   |
|------------|--------------|------------------------|-----------|---------------|
| $\gamma_E$    | normal       | $(3.00\text{ days})^{-1}$ | 0.1       | (0.6,1.4)     |
| $\gamma_A$    | normal       | $(2.50\text{ days})^{-1}$ | 0.1       | (0.6,1.4)     |
| $\gamma_1$    | normal       | $(3.00\text{ days})^{-1}$ | 0.1       | (0.6,1.4)     |
| $\gamma_2$    | normal       | $(7.25\text{ days})^{-1}$ | 0.1       | (0.6,1.4)     |
| $\gamma_3$    | normal       | $(7.25\text{ days})^{-1}$ | 0.1       | (0.6,1.4)     |
| $\beta_i$     | lognormal    | (see caption)          | 0.5       | (0.1,10)      |
| $\nu_L$       | lognormal    | 0.1                    | 0.5       | (0.1,10)      |
| $t_{lock}$    | normal       | 17 days                | 0.06      | (0.06,1.8)    |
| $W_{lock}$    | normal       | 12 days                | 0.08      | (0.008,1.7)   |
| $a_i^F$       | lognormal    | 0.2                    | 0.5       | (0.01,10)     |
| $r$          | normal       | 0.2                    | 0.1       | (0.05,0.5)    |

## D.2. Parameters, priors and initial conditions

This section gives additional details of parameters in the model of §6, and the priors used for inference. The infectiousness parameters $\nu_k$ are defined relative to the first infectious stage, so $\nu_1 = 1$. (This does not lose any generality because the $\nu$ parameters only appear through the combination $\beta_i\nu_k$.) In practice, we parametrize $\nu$ in terms of a single inference parameter $\nu_L$: we take $\nu_0 = \nu_1 = 1$, with $\nu_2 = \nu_3 = \nu_L$. The early stages of a COVID-19 case are much more infectious than later stages so $\nu_L < 1$. As discussed in §6, a common modelling assumption [8,10–12] is that the infectious period is (approximately) independent of whether the individual is symptomatic or not, which corresponds to $\nu_L = 0$. This approach is supported by some medical data [61,62]. For this illustrative computation, we retain $\nu_L$ as a parameter: its prior mean is 0.1 which means that most infections do happen in the early stages of the disease. The $\gamma$ and $\nu$ parameters are assumed independent of age, while $\beta$, $\alpha$, $f$, $a$ are age-dependent.

The fixed (age-dependent) CFR was estimated from the data for 18–24 March in table 4 of [60], by fitting an exponential dependence for ages 40–90 and then extrapolating this function to younger age groups. This leads to $f = Ae^{(\text{age})/\xi}$ with $A = 1.43 \times 10^{-4}$ and $\xi = 13.1$ years. The fraction of asymptomatic/paucisymptomatic cases is taken from table 2 of [60], using linear interpolation to obtain values for the cohorts considered here. The relevant numbers are shown in figure 14. As discussed in the main text, these numbers are subject to considerable uncertainty, but the resulting model is flexible enough to fit the data used here. It would be valuable to incorporate additional data, to constrain these variables, for example through testing data, which can provide information on the number of cases.

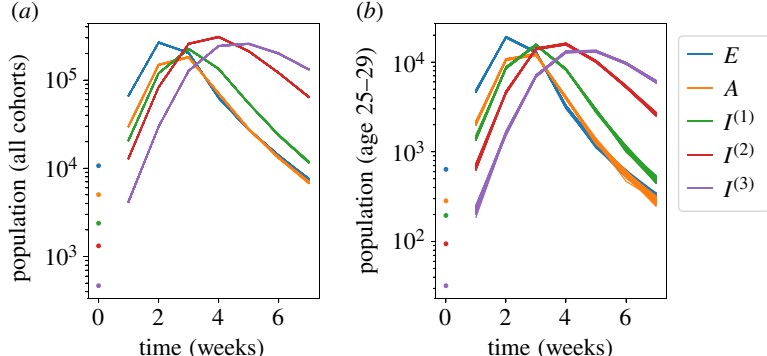

**Figure 15.** Sampling of latent variables conditional on the data, to illustrate the size of the fluctuations described by the functional CLT. (a) Total population in each epidemiological class, as obtained from a conditional nowcast with 100 trajectories, using MAP parameters. The dots indicate the inferred (MAP) initial condition. ($C^F$ model variant with step-like-NPI). (b) Population of latent compartments for a single cohort (age 25–29), the stochastic fluctuations more apparent at this level.

The priors for inferred epidemiological parameters are summarized in table 1. The $\gamma$ parameters are fixed by the disease itself and can be constrained based on medical data, see e.g. [71] for a discussion. We take Gaussian priors for these parameters with standard deviation 10% of the mean. Other parameters like $\beta_i$, $\nu_L$, $a_i$, $r$ are associated with disease transmission, and are much less well characterized, hence the use of less informative priors, which are lognormal with standard deviation 50% of the mean. (For positive parameters with large uncertainty, the lognormal prior is much more heavy-tailed than a Gaussian with the same standard deviation, while still penalizing very small values.) The lockdown parameters $t_{lock}$ and $W_{lock}$ have a prior standard deviation of 1 day.

To determine the initial condition for the model in terms of the inference parameters, we use the linearized dynamics to obtain (C 1). As a first estimate, it is reasonable to take $x(0) = x_{lin}$ with $\kappa$ taken as an inference parameter. In practice, our initial condition is obtained by modifying this $x_{lin}$. First, the $D$ (deceased) compartments are initialized from the observation data, which overrides the value from the dominant eigenvector. Second, the $E$, $A$, $I^{(k)}$ compartments for the oldest cohort are determined by a separate procedure (detailed in the next paragraph), which allows extra flexibility in the inference. Finally, the $S$ compartments (for all age cohorts) are chosen to enforce the (fixed) total population of each cohort. (Using (C 1) automatically ensures the correct cohort populations, but modifying $x(0)$ from $x_{lin}(0)$ means that compensation is required, to enforce this constraint.)

The modified initial condition for the oldest cohort is based on a hypothesis that infections started in younger age groups, before spreading into the elderly population. The inferred result is consistent with such a hypothesis. Since initial conditions are unknown *a priori*, we take broad prior distributions, which are lognormal with standard deviation one half of the mean. The prior mean values were chosen based on preliminary computations, to obtain reasonable agreement with deaths in the first few weeks. Denoting these mean values by $\mu$ we take $\mu_\kappa = 5 \times 10^{-4}$ and for the oldest cohort $(\mu_E, \mu_A, \mu_{I^1}, \mu_{I^2}, \mu_{I^3}) = (2000, 1200, 300, 60, 40)$, see also figure 16 below.

## D.3. Contact matrices

The contact matrices for $C^P$ and $C^F$ model variants are based on [63,64], as we now explain. On general grounds, one expects contacts to obey a *reciprocal relation*: the matrix $Q$ with elements

$$Q_{ij} = N_i C_{ij}, \qquad (D\ 2)$$

should be symmetric, $Q_{ij} = Q_{ji}$.

For $C^P$, the contact matrix is taken from [63], by summing the contributions from home/work/school/other. It is notable (e.g. figure 3) that the data do not satisfy (D 2), this can be traced back to the reporting of contacts by the participants of [65].

For $C^F$, the data in [64] are provided as a (non-normalized) estimate of $Q$, based on $n_Q$ single-year age cohorts. As for $C^P$, we sum the contributions from different environments to obtain a single matrix. Let $\mathcal{A}_i$

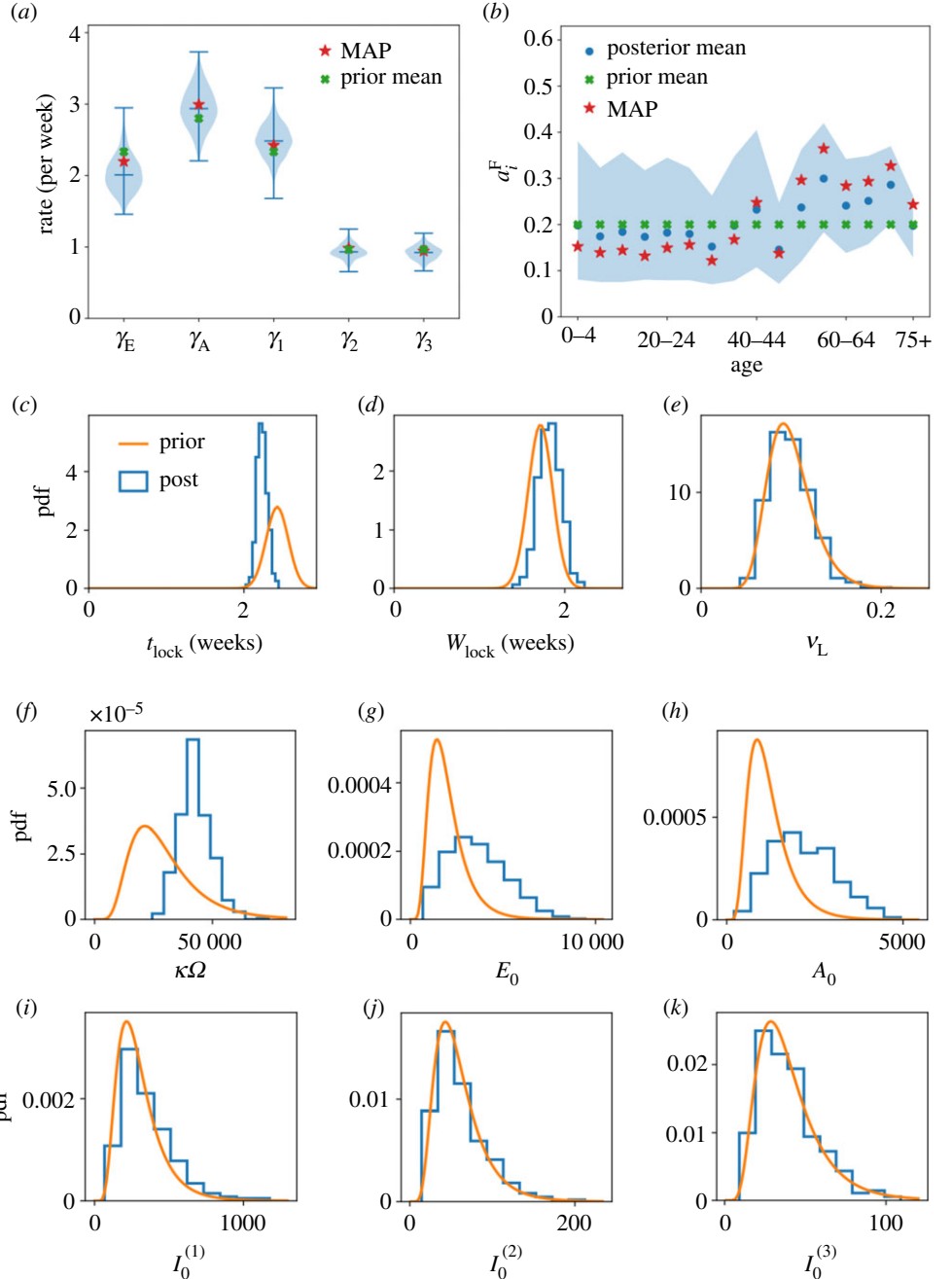

**Figure 16.** Posterior for parameters, $C^F$ model variant with step-like-NPI. (a) $\gamma$ parameters, the violin plots are based on kernel density estimates. (b) $a_i^F$ parameters, the shading is 5th to 95th percentile of the posterior. (c–e) Prior and posterior distributions of lockdown parameters $t_{lock}$, $W_{lock}$ and late-stage infectivity parameter $v_L$. (f–k) Prior and posterior distributions of initial condition parameters, specifically, the coefficient $\kappa$ of the leading mode, and the individual compartment populations for the oldest (75+) cohort.

be the set of single-year age cohorts corresponding to the (5-year) cohort $i$, and denote the reported estimate of $Q$ by $Q^0$. Then we take

$$C_{ij} = \frac{\Omega}{\chi N_i} \sum_{p \in \mathcal{A}_i} \sum_{q \in \mathcal{A}_j} Q^0_{pq}. \tag{D 3}$$

The constant $\chi$ is included because $Q^0$ is not normalized, we take $\chi = 3Mn_Q$ so that the typical numbers of contacts are comparable with [63]. This scaling is somewhat arbitrary but errors/uncertainties in this factor can be compensated by rescaling the $\beta$ parameters of the model. The matrix $Q^0$ is symmetric so the resulting contact matrices obey (D 2).

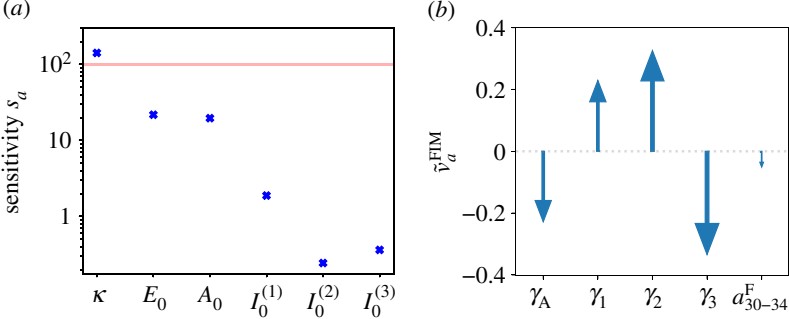

**Figure 17.** (*a*) Sensitivities for the parameters determining initial conditions. (*b*) Soft mode of the FIM, showing the five largest $\tilde{v}_a^{\text{FIM}}$ among the model parameters (we have excluded in this case parameters relevant for initial conditions). This mode illustrates that the model behaviour (and hence the likelihood) is almost unchanged if one increases the parameters $\gamma_1$, $\gamma_2$ and simultaneously reduces $\gamma_A$, $\gamma_3$. Other model parameters have small contributions to this mode, as illustrated by the $a_i^{\text{F}}$ parameter, which is the next largest element in magnitude.

## D.4. Additional results

This section shows additional results from the inference methodology.

We have emphasized that our Bayesian analysis accounts for all sources of uncertainty in the model, including parameter uncertainty, and the stochasticity inherent in the compartment model. As a direct measure of this stochasticity, figure 15 shows a conditional nowcast, as defined in §4.4. We show results summed over age cohorts, and for one representative cohort. These results illustrate the fluctuations that are captured by the functional CLT. For the summed data, the fluctuations are small, consistent with the relatively large numbers of individuals. At the level of specific cohorts, the fluctuations are significant.

Figure 16 shows the posterior distributions of parameters for the $C^{\text{F}}$ model with step-like NPI. This complements figure 6*b* of the main text where similar results are shown for the parameter $\beta$. In most cases, the posterior distributions overlap strongly with the priors. We note, however, that the parameters have significant correlations under the posterior, which are not apparent here since we only show the marginals for individual variables. (For example, the initial rate of epidemic growth depends on a particular combination of the $\gamma$ and $\beta$ parameters; this growth rate is tightly constrained by the data, even if individual $\gamma$ and $\beta$ parameters remain uncertain.)

For the $C^{\text{F}}$ model variant with step-like-NPI, we evaluated the FIM at the MAP parameters. Hence, (i) we gain understanding of how sensitive our model is expected to be to small parameter perturbations, and (ii) we understand whether there are soft directions along which the likelihood depends weakly on the parameters. The sensitivities for model parameters are discussed in §7.2, see figure 7. There are corresponding sensitivities for the parameters that determine the initial condition, see figure 17*a*. The parameter $\kappa$ is sensitive to the data, as expected since it determines the size of the epidemic at early times.

The soft modes of the likelihood are determined by the eigenvalues and eigenvectors of the FIM. The eigenvectors corresponding to small eigenvalues define directions in which the likelihood is expected to change very little. Let $v^{\text{FIM}}$ be an eigenvector of the FIM $\mathcal{I}$ with a small eigenvalue—its elements correspond to differences of the parameters $\boldsymbol{\theta}$ from their MAP values. It is convenient to normalize these as fractional changes with respect to the MAP by defining a vector $\tilde{v}^{\text{FIM}}$ with elements

$$\tilde{v}_a^{\text{FIM}} = \frac{1}{\theta_a^*} v_a^{\text{FIM}}, \tag{D 4}$$

which is normalized such that $\sum_a |\tilde{v}_a^{\text{FIM}}|^2 = 1$. Hence large values of $\tilde{v}_a^{\text{FIM}}$ indicate parameters that are significantly affected by the soft mode.

An example is given in figure 17*b*. If one increases $\gamma_A$ and reduces $\gamma_1$ appropriately, the result is a model with the same (mean) infectious period. A similar effect is obtained by increasing $\gamma_2$ and reducing $\gamma_3$. Due to these degenerate directions in the parameter space, a pure maximum-likelihood estimation (MLE) approach to inference of any model displaying soft modes potentially leads to wrong results. Bayesian inference on the other hand has the natural ability to remove soft modes by virtue of the additional information provided by priors.

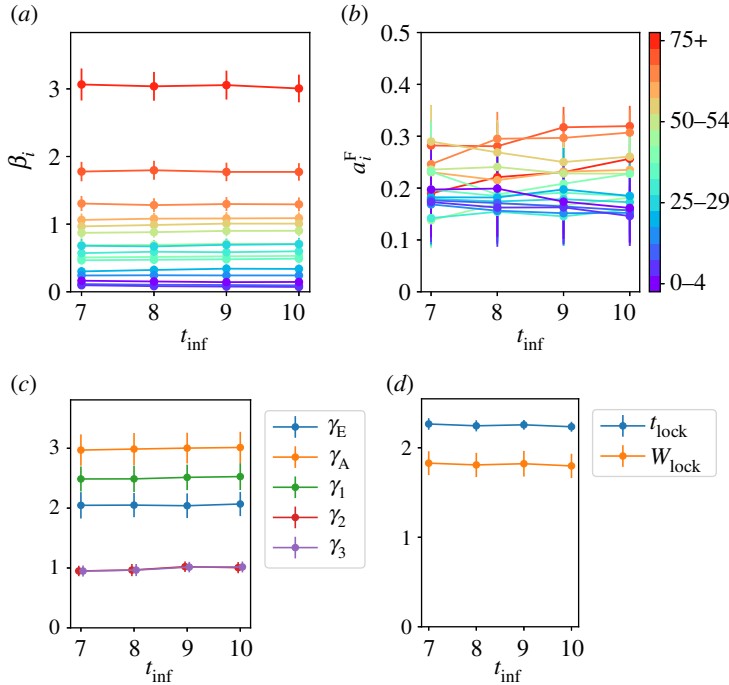

**Figure 18.** Dependence of inferred parameters on time used for inference, see also figure 4. Plots show posterior mean with error bars showing posterior standard deviation. ($C^F$ model variant, NPI-with-easing.)

Finally, we consider the effects of the inference window. To complement figures 10 and 11, we show in figure 18 the dependence of inferred parameters on the time period used for inference (figure 11 shows similar results for the parameters $r$, $v_L$). Most parameters depend weakly on the time window, which indicates a robust forecast.

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
